# ToolChain*: Efficient Action Space Navigation in Large Language Models with A* Search

**Yuchen Zhuang**[1][*], **Xiang Chen**[2], **Tong Yu**[2], **Saayan Mitra**[2]
**Victor Bursztyn**[2], **Ryan A. Rossi**[2], **Somdeb Sarkhel**[2], **Chao Zhang**[1]
Georgia Institute of Technology[1] Adobe Research[2]
`yczhuang@gatech.edu`, `{xiangche, tyu, smitra}@adobe.com`
`{soaresbu, ryrossi, sarkhel}@adobe.com`, `chaozhang@gatech.edu`

## Abstract

Large language models (LLMs) have demonstrated powerful decision-making and planning capabilities in solving complicated real-world problems. LLM-based autonomous agents can interact with diverse tools (e.g., functional APIs) and generate solution plans that execute a series of API function calls in a step-by-step manner. The multitude of candidate API function calls significantly expands the action space, amplifying the critical need for efficient action space navigation. However, existing methods either struggle with unidirectional exploration in expansive action spaces, trapped into a locally optimal solution, or suffer from exhaustively traversing all potential actions, causing inefficient navigation. To address these issues, we propose ToolChain*, an efficient tree search-based planning algorithm for LLM-based agents. It formulates the entire action space as a decision tree, where each node represents a possible API function call involved in a solution plan. By incorporating the A* search algorithm with task-specific cost function design, it efficiently prunes high-cost branches that may involve incorrect actions, identifying the most low-cost valid path as the solution. Extensive experiments on multiple tool-use and reasoning tasks demonstrate that ToolChain* efficiently balances exploration and exploitation within an expansive action space. It outperforms state-of-the-art baselines on planning and reasoning tasks by 3.1% and 3.5% on average while requiring 7.35x and 2.31x less time, respectively.

## 1 Introduction

Large language models (LLMs), such as GPT (Radford et al., 2018; 2019; Brown et al., 2020; OpenAI, 2023) and PaLM (Chowdhery et al., 2022; Anil et al., 2023), have exhibited remarkable capabilities of reasoning and instruction-following across a wide range of tasks (Huang & Chang, 2023). Recently, instructing LLMs to utilize external tools for complex real-world problems has emerged as a topic of growing importance (Hao et al., 2023b; Zhang et al., 2023; Zhuang et al., 2023; Yang et al., 2023b; Schick et al., 2023; Lu et al., 2023). For complicated tasks, LLM-based autonomous agents integrate LLMs with various external tools (APIs), generating solutions that involve intermediate reasoning steps (Schick et al., 2023; Lu et al., 2023; Patil et al., 2023; Qin et al., 2023b). Given a problem description, the goal of an agent is to determine a chain of API function calls that can be executed sequentially toward a valid solution. However, given an action space of hundreds of candidate API functions, each comprised of various function names and parameters available at every planning step, searching for a globally optimal solution becomes highly challenging.

Existing methods that leverage LLMs as autonomous agents for decision-making and reasoning can be broadly classified into four categories (Figure 1): (1) *open-loop methods* (Wei et al., 2022; Zhou et al., 2022; Huang et al., 2022a; Shen et al., 2023; Lu et al., 2023) generate a complete plan for problem-solving without any adaptation during the execution; (2) *greedy closed-loop methods* (Yao et al., 2023b; Jang, 2023; Huang et al., 2022b; Kim et al., 2023; Liang et al., 2022) leverage environmental feedback to greedily determine the next step in the plan; and (3) *closed-loop methods* (Wang et al., 2023; Sun et al., 2023) incorporate environment feedback to continuously monitor system behaviors

---

[*]Work done during the author's internship at Adobe Research.

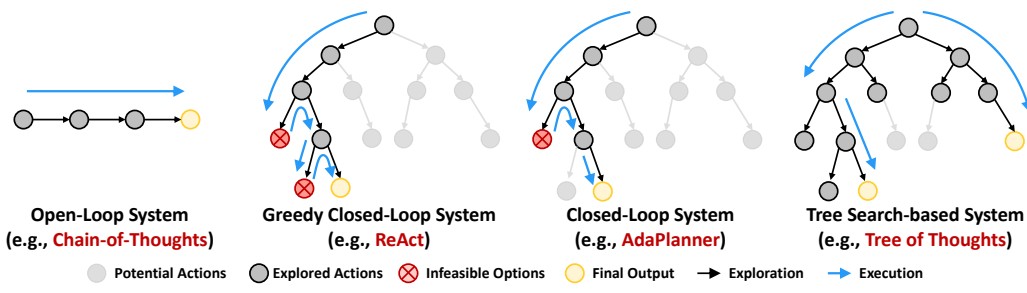

Figure 1: A comparison of existing methods that leverage LLMs for decision-making from a searching space perspective. Most existing methods of (1) open-loop systems (*e.g.*, Chain-of-Thought (Wei et al., 2022)), (2) greedy closed-loop systems (*e.g.*, ReAct (Yao et al., 2023b)), and (3) closed-loop systems (*e.g.*, Adaplanner (Sun et al., 2023)) only explore one possible direction. This often leads to limited exploration of the entire action space. In contrast, (4) tree search-based methods (*e.g.*, Tree-of-Thoughts (Yao et al., 2023a)) identify a valid solution path by extensively examining multiple decision space branches, covering almost every conceivable node. Our proposed ToolChain* belongs to the tree search-based category and improves by developing an efficient search algorithm.

and modify subsequent plans accordingly. However, such unidirectional navigation systems have two major limitations: **error propagation**, originating from a mistaken action and leading to a faulty loop; **limited exploration**, despite being equipped with plan refinement strategies, most existing methods only explore a small portion of the large action space, falling into locally optimal solutions. To this end, few studies initiate exploring (4) *tree search-based methods* (Yao et al., 2023a; Hao et al., 2023a) for leveraging multiple reasoning paths simultaneously and evaluating branches to decide the next course of action. However, existing tree search-based algorithms, such as depth-first search (DFS) (Yao et al., 2023a) and Monte Carlo Tree Search (MCTS) (Hao et al., 2023a), require exhaustive exploration of nearly all potential actions within the entire decision space, resulting in **inefficient searches** for globally optimal solutions.

To address these limitations, we propose ToolChain*, an efficient A* tree search-based planning method for LLM-based agents. We formulate the tool-use planning process as a decision tree, where each node represents a potential API call for a given step. Aligned with the traditional A* search algorithm, the proposed ToolChain* determines which paths to extend based on both the cost of the current path and an estimated future cost required for completing the current plan. With task-specific cost functions, erroneous actions will be penalized and mitigated, as these actions cause additional costs when propagated along the path, leading the path to be progressively de-prioritized and left unexpanded over iterations. In addition, unlike the simulation stage in MCTS, which requires multiple steps to simulate until a terminal state during rollout, the future cost estimation in ToolChain* enables expansion of only the next step. With efficient node expansion, ToolChain* effectively searches for globally optimal solutions within a manageable number of steps.

Our main contributions are as follows: (1) We propose ToolChain*, a novel A*-like tree search algorithm, to develop autonomous LLM-based agents for complex planning and reasoning tasks; (2) ToolChain* formulates the action space as a decision tree, effectively mitigating error propagation and expanding search space; and (3) ToolChain* significantly accelerates LLM-based agents in navigating expansive action tree spaces, striking a balance between exploring unvisited actions and exploiting global optimal solutions.

## 2 PRELIMINARIES

**Problem Formulation.** Leveraging LLMs as agents for problem solving can be conceptualized as a planning process. For initialization, the LLM agent is augmented with access to a pool of $m$ candidate API functions, denoted as $\mathcal{A} = \{\text{API}_0, \text{API}_1, \cdots, \text{API}_m\}$, along with a natural language task description $g \in \mathcal{G}$ from the task space $\mathcal{G}$. The objective of the LLM agent is to translate the task description $g$ into an ordered sequence of $T_g$ API function calls $p_g = \{a_0, a_1, \cdots, a_{T_g}\}$. Specifically, considering the task description $g$ as the initial state $s_0$, we sample the plan $p_g$ by prompting the LLM

agent with the API definitions $\mathcal{I}$ and demonstration samples $\mathcal{D}$ as: $p_g \sim \rho(a_0, a_1, \cdots, a_{T_g}|s_0; \mathcal{I}, \mathcal{D})$ : $\mathcal{G} \times \mathcal{I} \times \mathcal{D} \rightarrow \Delta(\mathcal{A}^{T_g})$, where $\Delta(\cdot)$ is a probability simplex function. The final output is derived after executing the entire plan $y \sim \pi(y|s_0, a_1, a_2, \cdots, a_{T_g})$, where $\pi(\cdot)$ indicates a plan executor.

**Tree Search-Based Systems.** Tree search methods frame a planning problem as a search over a decision tree, where each node $n$ represents an action $a_n$, accompanied by a state $s_n \in \mathcal{S}$ indicating a valid path from the initial state to the current action. When exploring the tree space, tree search approaches expand $k$ potential child nodes $ch(n)$ of the current node $n$ via sampling from the potential action set generated by LLMs $a_{ch(n)}^{(j)} \sim \rho(a_{ch(n)}|s_n; \mathcal{I}, \mathcal{D}), (j = 1, \cdots, k)$ and add the new nodes to the tree state space $\mathcal{S} = \mathcal{S} \cup \{(s_n, a_{ch(n)}^{(j)})\}_{j=1}^k$. With value functions for state evaluation, tree search-based methods aim to identify a path from the root node $s_0$ to the leaf nodes with the highest value or lowest cost. Our proposed ToolChain$^*$ is a tree search-based method.

**Monte Carlo Tree Search.** MCTS, which employs heuristic exploration to construct its search tree, has achieved great success in decision-making tasks, such as GO (Silver et al., 2016). Its variant, UCT (Kocsis & Szepesvári, 2006), has been adopted in Hao et al. (2023a) for the development of LLM-based agents. Specifically, it initiates from the root node of the task description $g$ and moves down the tree by selecting optimal actions (child nodes) until the leaf node. Then, MCTS introduces one or multiple child nodes based on available actions provided by LLMs and identifies the most promising node $n$. From the newly expanded node $n$, MCTS requires LLM agents to execute a simulated rollout until a terminal state is reached. Upon completing the simulation, a result is returned from $n$ all the way back to the root node, accompanied by the value function $Q(n)$ to update all the scores on the selected path.

**MCTS vs. A$^*$ Search.** Despite the performance gains attained by MCTS in planning and reasoning tasks, its direct application to LLM agents comes with significant execution costs. The rollout mechanism within MCTS requires multiple LLM calls to prompt the next actions until a terminal state. Furthermore, unlike two-player zero-sum games, the planning tasks essentially operate as one-player games, where value functions estimated by random rollouts might exhibit significant inaccuracies. To mitigate the issue, ToolChain$^*$ is proposed based on a more efficient A$^*$ search algorithm. A comparison between MCTS and our proposed ToolChain$^*$ is illustrated in Figure 2. Unlike MCTS, A$^*$ search

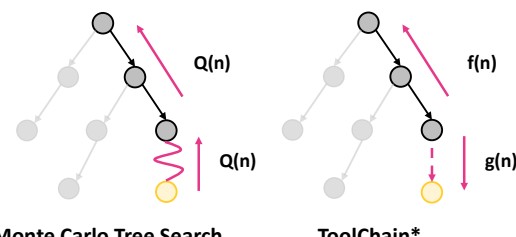

Monte Carlo Tree Search      ToolChain*

Figure 2: A comparison between MCTS and A$^*$ search in ToolChain$^*$. Unlike MCTS, A$^*$ search only requires one-step expansion guided by cost.

necessitates only a single LLM call for determining the next actions during expansion according to two cost functions, $g(n)$, quantifying the cost of the path from the root node to $n$, and $h(n)$, a heuristic function estimating the cost of the most promising path from $n$ to the goal.

## 3   TOOLCHAIN$^*$: A TREE SEARCH PERSPECTIVE ON EXTERNAL TOOL USE

In this section, we introduce the ToolChain$^*$ that enables LLM-based agents to efficiently navigate the action space to identify a valid solution path for problem-solving (Figure 3). First, we outline the framework of ToolChain$^*$ (Section 3.1), consisting of three iterative stages: *selecting* the most promising path in the explored decision tree, *expanding* the potential following actions along the selected path, and subsequently *updating* the cost functions. Within ToolChain$^*$, the cost function is composed of two components: cumulative cost $g(n)$ (Section 3.2) and future score $h(n)$ (Section 3.3).

### 3.1   OVERVIEW

ToolChain$^*$ is a best-first search algorithm, efficiently guiding LLM agents in generating a sequence of API function calls as a solution plan. We formulate the action space as a search tree $\mathcal{T}$, where each node $n$ represents an action $a_n$, accompanied by a state composed of the initial task description $s_0$ and previous actions. This facilitates the translation of action sequence planning into a navigation

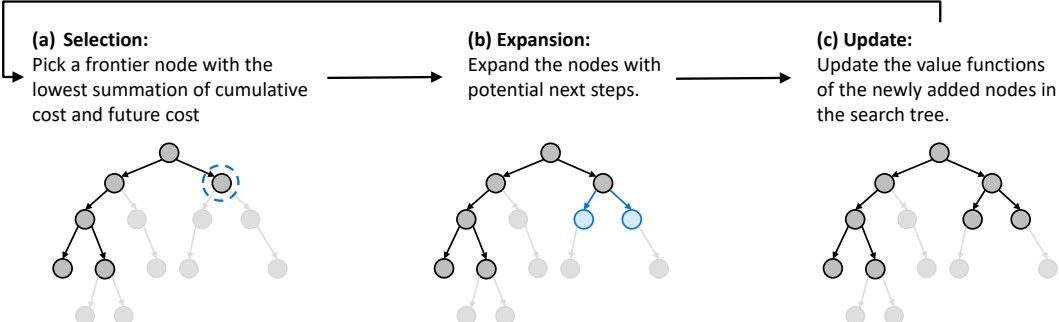

Figure 3: ToolChain* framework of three phases: (a) **selection**, (b) **expansion**, and (c) **update**. The dark and grey circles indicate the explored actions and the potential but unexplored ones, respectively. The blue circles represent the selected next step.

task originating from the root node of the decision tree. ToolChain* starts the search tree $\mathcal{T}$ with a single root node, corresponding to the input input problem description $s_0$. At each step, it selects a node $n$ from the frontiers of $\mathcal{T}$ (denoted as $\mathcal{F}(\mathcal{T})$) according to the cost function. Then, it expands $n$ with the LLM to generate a set of $k$ potential i.i.d. actions $\{a_{ch(n)}^{(j)}\}_{j=1}^{k}$ for the next step and grows $\mathcal{T}$ with the generated actions. Finally, we update the actions into new nodes $s_{ch(n)}^{(j)} = (s_n, a_{ch(n)}^{(j)})$ and update their cost functions accordingly. Algorithm 1 describes the procedure in detail.

**Selection.** Given a search tree $\mathcal{T}$, we denote its nodes as $\mathcal{V}(\mathcal{T})$. The frontier $\mathcal{F}(\mathcal{T}) \subseteq \mathcal{V}(\mathcal{T})$ contains all the leaf nodes in $\mathcal{T}$ that have yet to be explored. Given our objective to minimize the total cost of the final solution, the optimal next node to expand would be the most promising plan as part of the best solution. Assume we possess a cost function oracle $f(n)$, which provides the cost of the best plan incorporating $n$ to address the problem $s_0$ under $\mathcal{T}$. Then, we can select the next node with the lowest cost: $n_{next} = \arg\min_{n \in \mathcal{F}(\mathcal{T})} f(n)$. A proper design of the value function $f(n)$ not only augments search efficiency but also aids in identifying globally optimal solutions.

**Expansion.** Once the node $n$ with the minimum cost estimation $f(n)$ has been selected, we expand the search tree with $k$ potential actions for the next step. These actions are sampled from the potential

---

**Algorithm 1:** ToolChain*.

**Input:** $x$: input; $\rho$: large language model; $T$: the maximum exploring steps; $\mathcal{T}$: the decision tree; $\mathcal{F}(\mathcal{T})$: the set of frontier nodes in $\mathcal{T}$; $f(n)$: the cost function of node $n$.

Initialize $\mathcal{T} = \{\mathcal{V}, \mathcal{E}\}, \mathcal{V} \leftarrow x, \mathcal{E} \leftarrow \varnothing$
**for** $t = 1, 2, \cdots, T$ **do**
$\quad n_{next} \leftarrow \arg\min_{n \in \mathcal{F}(\mathcal{T})} f(n)$ // *Selection*
$\quad \{a^{(i)}\}_{i=1}^{k} \leftarrow \rho(n_{next})$ // *Expansion*
$\quad$ **for** $i = 1, 2, \cdots, k$ **do**
$\quad\quad$ Add $[n_{next}, a^{(i)}]$ to $\mathcal{T}$ under $n_{next}$
$\quad$ Update $f(n)$ for $n$ in $\mathcal{F}(\mathcal{T})$. // *Update*

**Output:** The valid path to solve the problem $\arg\max_{n \in \mathcal{F}(\mathcal{T})} f(n)$.

---

action set generated by LLMs $a_{ch(n)}^{(j)} \sim \rho(a_{ch(n)}|s_n; \mathcal{I}, \mathcal{D}), (j = 1, \cdots, k)$, given the API definitions $\mathcal{I}$ and demonstration examples $\mathcal{D}$. For the generated actions or reasoning steps $\{a_{ch(n)}^{(j)}\}_{j=1}^{k}$, we establish their corresponding nodes under node $n$. Contrasting with the approach in MCTS (Hao et al., 2023a), which requires multiple calls to $\rho$ until a terminal state during rollout, our expansion only requires a single call to generate the possible actions at the next step.

**Update.** Denote the search tree $\mathcal{T}$ after expansion of node $n$ as $\mathcal{T}'$. Given that new nodes have been incorporated and the original tree structure has changed, we need to update the frontier nodes as $\mathcal{F}(\mathcal{T}')$. With the new frontier nodes $n \in \mathcal{F}(\mathcal{T}')$, we can compute their corresponding cost functions for the next selection-expansion-update iteration.

**Cost Function.** We draw inspiration from A* algorithm to design and update the cost function $f(n)$. Specifically, A* selects the path that minimizes $f(n) = g(n) + h(n)$, where $n$ is the current node, $g(n)$ represents the cost of the path from the start node to $n$, and $h(n)$ is a heuristic function estimating the cost of the cheapest path from $n$ to the goal.

## 3.2 Design of Cumulative Cost $g(n)$

During the planning process, we assess the cumulative cost of actions in the current plan and guide the planning based on the assessment. For each node $n$ in the searching tree, we design a single-step value function $g_t(n)$ ranging from 0 to 1 and formulate the cost as its complement $1 - g_t(n)$. Thus, the cumulative cost of $n$ can be computed by summing up all the single-step costs of its ancestor nodes $an(n)$: $g(n) = \sum_{i \in an(n)} 1 - g_t(i)$. More specifically, we combine two different value functions, the task-specific heuristic function from reference data (long-term memory) $g_{t,1}(n)$ and the self-consistency frequency by LLM $g_{t,2}(n)$, to compute cumulative cost $g(n)$:

$$g(n) = \sum_{i \in \{an(n),n\}} (1 - g_{t,1}(i))^\alpha \cdot (1 - g_{t,2}(i))^{1-\alpha}, \tag{1}$$

where $\alpha$ is a weight parameter for the geometric mean.

**Task-Specific Heuristic Function $g_{t,1}(n)$.** We can also maintain a long-term memory with successful experiences and compute a heuristic score accordingly. The long-term memory starts from a seed set of demonstration examples provided in a specific dataset and is iteratively extended with successful plans during evaluation. Each example within the long-term memory is represented as a plan $m_j = (s_{j,0}, a_{j,1}, a_{j,2}, \cdots, a_{j,T_j}) \in \mathcal{M}$. The number of actions $T_j$ in the plan varies case-by-case. To leverage the successful experiences for evaluating the current plan, we compute the longest common sub-sequence (LCS) score between the current generated plan $s_n$ and each plan $m_j$ in the long-term memory $\text{LCS\_score}(s_n, m_j) = \frac{\text{LCS}(s_n, m_j)}{\min(L(s_n), L(m_j))}$, where $L(\cdot)$ indicates the length of the plan. Following this, we compute the cumulative functions as the highest LCS score $g_{t,1}(n) = \max_{m_j \in \mathcal{M}} \text{LCS\_score}(s_n, m_j)$, measuring the proportion of success in the plan relative to the experiences accumulated in the long-term memory.

**Self-consistency Frequency $g_{t,2}(n)$.** Self-consistency (Wang et al., 2022b) is an ensemble approach that samples $k$ i.i.d. actions at the next step $\{a_{t+1}^{(j)}\}_{j=1}^k \sim p(a_{t+1}|x, a_0, a_1, \cdots, a_t)$. We then select the semantically different actions from the $k$ generated samples as the set of potential next steps. For tool-use scenarios, as the actions are strict in format of API functions and parameters, we directly construct the set with non-repeating actions. For reasoning scenarios, however, actions represent intermediate thought processes articulated in natural language. Inspired by Kuhn et al. (2022), we apply a DeBERTa-large model (He et al., 2020) fine-tuned on natural language inference (NLI) dataset MNLI (Williams et al., 2018) to determine whether the two generated actions entail each other semantically. This allows us to discard actions that are semantically equivalent, only retaining those that offer distinct reasoning as potential next steps. Lastly, we consider the frequencies of different actions in the set as their corresponding cumulative score, given by $g_{t,2}(n) = \#\{j|a_{t+1}^{(j)} = n\}/k$.

## 3.3 Design of Future Cost $h(n)$

Similar to the formulation of cumulative cost $g(n)$, we integrate two distinct reward functions, the task-specific heuristic function $h_{t,1}(n)$ and the Imagination Score by LLM $h_{t,2}(n)$, to compute $h(n)$:

$$h(n) = (1 - h_{t,1}(n))^\beta \cdot (1 - h_{t,2}(n))^{1-\beta}, \tag{2}$$

where $\beta$ is the geometric mean weight for future cost.

**Task-Specific Heuristic Function.** Similar to the heuristic function in the cumulative cost (Section 3.2), we continue to leverage the long-term memory to compute the future score. From the long-term memory, we can derive the average relative position score of the action $a$ appearing in the plans $m_j$: $h_{t,1}(a) = \sum_{m_j \in \mathcal{M}} \mathbb{1}_{\{a \in m_j\}} \frac{pos(a, m_j)}{T_j}$, where $pos(a, m_j)$ indicates the relative position of action $a$ in the plan $m_j$. Note that the action space can be infinite, and the long-term memory may not cover all potential actions relevant to unseen tasks. Thus, given an action node $n$, we compute its future score as the heuristic score of the lexically closest action covered in the long-term memory: $h_{t,1}(n) = h_{t,1}(\arg\max_{a \in \mathcal{M}} \text{LCS\_score}(n, a))$.

**Imagination Score by LLM.** Directly querying LLMs for self-evaluation of the future cost at the current step often yields over-confident scores (Lin et al., 2022). To address this, we enable LLMs to imagine more concrete future steps until the target $n_T$. However, it is worth noting that the imagined

Table 1: Main experiment results (success rate) on ToolBench, including tool use scenarios of (1) Home Search, (2) Trip Booking, (3) Google Sheets, and (4) Virtual Home.

| Models | GPT-3.5-turbo | | | | | GPT-4 | | | | |
|---|---|---|---|---|---|---|---|---|---|---|
| | Home Search | Trip Booking | Google Sheets | Virtual Home | Average | Home Search | Trip Booking | Google Sheets | Virtual Home | Average |
| GPT (OpenAI, 2023) | 80.0 | 85.8 | 51.4 | 18.9 | 59.2 | 97.0 | 96.7 | 62.9 | 23.5 | 70.0 |
| ReAct (Yao et al., 2023b) | 83.0 | 86.7 | 47.1 | 20.5 | 59.3 | 94.0 | **97.5** | 64.3 | 22.7 | 69.6 |
| AdaPlanner (Sun et al., 2023) | 90.0 | 87.5 | 55.7 | 20.7 | 63.5 | 97.0 | **97.5** | 66.7 | 27.1 | 72.1 |
| ToT-DFS (Yao et al., 2023a) | 82.0 | 81.7 | 53.4 | 21.0 | 59.5 | 95.0 | 96.7 | 62.9 | 24.8 | 69.9 |
| ToT-BFS (T=5) (Yao et al., 2023a) | 83.0 | 83.3 | 48.6 | 21.8 | 59.9 | 92.0 | 94.2 | 64.3 | 26.6 | 69.3 |
| MCTS (Hao et al., 2023a) | 85.0 | 86.7 | **62.9** | 24.4 | 64.8 | 96.0 | 94.2 | 66.7 | 31.3 | 72.1 |
| ToolChain* | **93.0** | **90.8** | 61.4 | **28.6** | **68.5** | **98.0** | **97.5** | **68.6** | **34.5** | **74.7** |

actions may not align with the real executed actions in future plans. To this end, we compute the future score as the proportion of current steps present in the imagined plan, *i.e.*, the ratio of the number between the current node $n$ ancestors to the target node $n_T$: $h_{t,2}(n) = \frac{|\{an(n)\}|}{|\{an(n_T)\}|}$. A higher score suggests that the imagined plan closely captures the path to the current step, indicating that fewer remaining steps are needed to accomplish the task in the imagination of LLMs.

## 4 EXPERIMENTS

In this section, we demonstrate the effectiveness and efficiency of ToolChain* through comprehensive experiments across a wide range of tool-use scenarios from ToolBench (Xu et al., 2023) (Section 4.2). In addition, we conduct extensive experiments on GSM8K (Cobbe et al., 2021) (Section 4.3) to showcase the generalization of ToolChain* on pure reasoning tasks without tool interaction.

### 4.1 EXPERIMENTAL SETUP

**Datasets.** We evaluate ToolChain* on four tool-use environments in *ToolBench* (Xu et al., 2023) and one reasoning task in *GSM8K* (Cobbe et al., 2021). For tool-use scenarios, we select environments with both a vast action space comprising a large number of function tools, and a requirement of a deep solution path with multiple API functions (*i.e.*, complicated tasks), including **Home Search**, **Trip Booking**, **Google Sheets**, and **Virtual Home**. Given that numerical reasoning requires multi-step computations to calculate answers, we choose **GSM8K** (Cobbe et al., 2021) for evaluation on math reasoning. Dataset details are available in Appendix D.1.

**Baselines.** For environments from ToolBench, we compare ToolChain* with the state-of-the-art LLM planning algorithms from three main categories, including *open-loop systems* (**GPT** (OpenAI, 2023)), *closed-loop systems* (**ReAct** (Yao et al., 2023b) and **Adaplanner** (Sun et al., 2023)), and *tree search-based systems* (**Tree-of-Thoughts** (Yao et al., 2023a) and **MCTS** (Hao et al., 2023a)). For mathematical reasoning problems, we employ a similar set of baselines as in the tool-use tasks. However, we exclude ReAct and AdaPlanner from mathematical reasoning evaluations. This is because they heavily depend on high-quality environment feedback to adjust action plans, which is unavailable in the GSM8K dataset. Additionally, since the action steps in the tool-use scenarios inherently form coherent sequences, we limit our comparison of ToolChain* to **Chain-of-Thought** (Wei et al., 2022) and **Self-Consistency** (Wang et al., 2022b) only for the math reasoning task, and exclude it from the ToolBench evaluations. Baseline details can be found in Appendix D.2.

### 4.2 TOOL USE: TOOLBENCH

We conduct experiments across four distinct planning tasks to assess the effectiveness and efficiency of ToolChain* in tool usage. The objective is to generate a sequence of API function calls to formulate a solution plan for each given task. For instance, these tasks include questions or requirements from users, *e.g.*, "*Could you help me find train tickets to Cape Coral?*". We present the main results, visualize the case study, analyze time-wise efficiency, and discuss ablation studies within the tool-use scenarios as follows. We report the success rate as the evaluation metric. Detailed task setup for ToolBench is available in Appendix B.3.

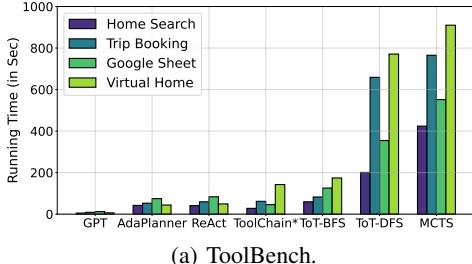
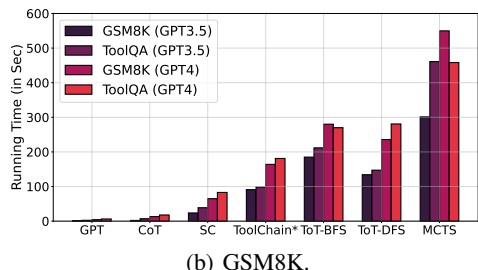

(a) ToolBench.                (b) GSM8K.

Figure 5: Time efficiency evaluation on (a) ToolBench and (b) GSM8K. We report the average running time in seconds over all instances in the dataset. ToolChain* achieves competitive efficiency to closed-loop systems without a tree structure and outpaces other tree search-based algorithms.

**Results.** Table 1 presents the main experiment results on ToolBench. Our proposed ToolChain* consistently outperforms across nearly all datasets, surpassing state-of-the-art baselines by margins of 3.7% and 2.5% with the base LLMs GPT-3.5-turbo and GPT-4, respectively. In comparison with the strongest closed-loop baseline AdaPlanner, ToolChain* improves the average success rate by 3.8%. This improvement is because AdaPlanner relies heavily on environmental feedback, which may not always be available in the tool-use scenarios. Without such high-quality feedback, closed-loop methods tend to explore a restricted trajectory within the action space, making them more susceptible to propagating errors from previous actions to future plans.

Moreover, ToolChain* not only surpasses the strongest tree search-based method, MCTS, but also shows the ability to exploit a better solution plan within the same exploration budgets. This is because our proposed task-specific cost function allows ToolChain* to prioritize the expansion of the most promising branches. Additional analysis is available in Appendix D.3.

**Case Study.** Figure 4 depicts an example of ToolChain* (GPT-4) and *ReAct* (Yao et al., 2023b) on a "*take shower*" task in Virtual Home dataset. According to the ground truth (green, "*shower*"), ToolChain* generates the correct action plan (blue, "*shower*") with an expanded search space, whereas the baseline searching method gets trapped in a locally optimal solution (red, "*soap*"). This suggests that by formulating and expanding upon a tree-based action space, ToolChain* is capable of effectively searching for the globally optimal solution in complex multi-step planning tasks.

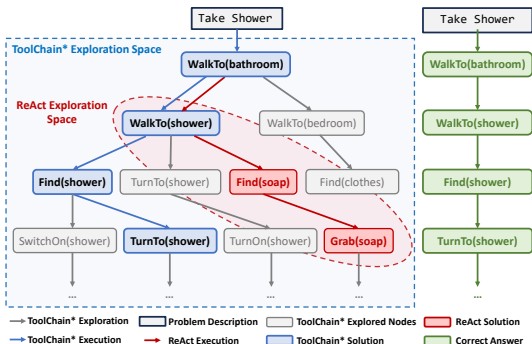

Figure 4: Case study of ToolChain* and *Re-Act* (Yao et al., 2023b) on Virtual Home dataset. Compared to ReAct with a unidirectional search (red), ToolChain* effectively enlarges search space (blue) with tree structures.

**Efficiency Evaluation.** In terms of efficiency, we evaluate the running time of ToolChain* against all the baselines based on GPT-3.5-turbo, as depicted in Figure 5(a). Remarkably, ToolChain* is 37.2% faster than the most efficient tree search-based method, Tree-of-Thoughts (BFS). This efficiency gain may stem from the proposed superior cost function, which efficiently navigates the most promising paths. Additionally, ToolChain* outpaces the best-performing tree search-based method, MCTS, by an impressive 415.84%. This discrepancy

Table 2: Ablation studies on ToolBench.

|  | Home Search | Trip Booking | Google Sheets | Virtual Home | Average |
|---|---|---|---|---|---|
| ToolChain* | 93.0 | 90.8 | 61.4 | 28.6 | 68.5 |
| $-g_{1,t}(n)$ | 91.0 | 88.3 | 60.0 | 22.6 | 65.5 |
| $-g_{2,t}(n)$ | 84.0 | 83.3 | 54.3 | 25.3 | 61.7 |
| $-h_{1,t}(n)$ | 88.0 | 87.5 | 61.4 | 23.0 | 65.0 |
| $-h_{2,t}(n)$ | 85.0 | 85.8 | 51.4 | 24.9 | 61.8 |
| $-g(n)$ | 61.0 | 34.9 | 44.2 | 21.0 | 40.3 |
| $-h(n)$ | 84.0 | 85.8 | 53.4 | 26.1 | 62.3 |

arises because ToolChain* focuses on expanding only the immediate next action during exploration. In contrast, MCTS goes through a more exhaustive process, simulating the entire future plan step by step using a rollout mechanism. Efficiency results based on GPT-4 are available in Appendix D.5.

**Ablation Studies.** We conduct ablation studies to evaluate the effectiveness (success rate) of both the cumulative and future cost functions (Table 2). The results suggest that each component of the cumulative and future cost functions contributes to the performance of ToolChain*. This verifies the efficacy of our proposed cost functions in guiding the search through the decision tree. In addition, eliminating either the entire cumulative or future cost results in a marked decline in the success rate. Relying exclusively on the future cost results in a sharp performance drop of 28.2%, deteriorating ToolChain* to a greedy strategy that favors the shortest solution plans with the least number of actions. Conversely, if the search is guided only by the cumulative cost, ToolChain* essentially mirrors the behavior of the BFS algorithm, yielding similar performance. Further analysis is in Appendix D.6.

### 4.3 MATH REASONING: GSM8K

Beyond tool-use scenarios, we demonstrate the flexibility of ToolChain* by generalizing its application to mathematical reasoning for solving math word problems. We conduct experiments on the entire set of GSM8K and also a subset of hard questions from GSM8K collected in ToolQA (Zhuang et al., 2023). Detailed task setup for GSM8K is available in Appendix B.4.

**Results.** Table 3 presents the main experimental results (accuracy) for GSM8K and its challenging subset from ToolQA. Similar to tool-use studies (Table 1), ToolChain* consistently outperforms all baselines in both the original set and the challenging subset. These results demonstrate the flexibility and generalization capabilities of ToolChain* in mathematical reasoning tasks. Notably, ToolChain* demonstrates greater advantages over other baselines on ToolQA (hard questions) than on GSM8K, indicating its superior capability in solving complicated tasks. This is because simpler questions are composed of simple and static reasoning, eliminating the need for multiple branches. In

Table 3: Main results on math reasoning task in GSM8K and its hard subset collected in ToolQA.

| Models | GPT-3.5-turbo | | GPT-4 | |
|---|---|---|---|---|
| | GSM8K | ToolQA | GSM8K | ToolQA |
| GPT | 67.3 | 26.0 | 86.6 | 66.0 |
| CoT | 70.1 | 30.0 | 87.5 | 75.0 |
| Self-Consistency | 76.1 | 47.0 | 92.4 | 78.0 |
| ToT-DFS | 69.9 | 32.0 | 89.2 | 76.0 |
| ToT-BFS | 72.3 | 39.0 | 91.3 | 77.0 |
| MCTS | 74.7 | 27.0 | 91.0 | 74.0 |
| ToolChain* | **77.0** | **52.0** | **93.5** | **84.0** |

contrast, challenging questions often involve complex reasoning, numerous intermediate steps, and multiple solution paths. The superior performance on hard subsets emphasizes the capability of ToolChain* in solving complicated reasoning problems. Furthermore, the efficiency analysis presented in Figure 5(b) indicates that ToolChain* ranks among the most efficient tree-based search baselines and has a time efficiency comparable to closed-loop systems without a tree structure. Detailed **case studies** of action space exploration and **efficiency analysis** with the number of valid actions are available in Appendix D.4 and D.5, respectively.

### 4.4 DISCUSSION: EMPIRICAL ANALYSIS

From the comprehensive evaluations in planning and reasoning tasks presented in Sections 4.2 and 4.3, we validate that ToolChain* addresses the two core limitations of open-/closed-loop LLM-based agents, error propagation in multi-step solutions and constrained exploration in expansive action spaces. Meanwhile, we demonstrate ToolChain* a more efficient searching strategy compared to existing tree search-based agents. From the scaling-up analysis in Figure 10 in Appendix D.5, alongside experimental results in Table 1 and efficiency metrics in Figure 5, we identify **a crucial trade-off** between effectiveness and efficiency in the direct application of tree search-based reasoning methods to complex tool use scenarios. To validate ToolChain* in solving these issues, we summarize key findings from experiments as follows: (1) From the main experimental results shown in Tables 1 and 3, ToolChain* surpasses open-/closed-loop and tree search

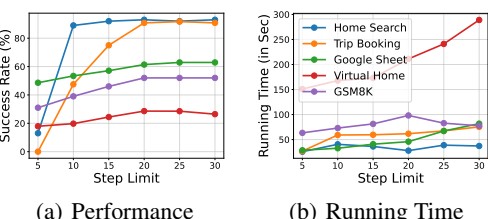

(a) Performance      (b) Running Time

Figure 6: Scaling analysis of ToolChain*. (a) Performance and (b) running time on ToolBench and GSM8K when scaling up step limitations $T$.

baselines in complex multi-step planning and reasoning tasks, effectively **mitigating error propagation**. A visualization example of how ToolChain* gradually abandons the faulty path and mitigates error propagation is available in Figure 7 in Appendix D.4. (2) From case studies in Figures 4, 7, and 8, ToolChain* navigates the path toward an optimal solution by formulating the action space as a decision tree, thereby extensively **broadening the exploration space**. (3) From Figures 5 and 9, ToolChain* significantly accelerates the search process compared to other tree search-based methods, **achieving time efficiency** even comparable to closed-loop systems without a tree structure. (4) From tool-use in ToolBench to math problems in GSM8K, we show that ToolChain* is a **plug-and-play generalizable framework** applicable to a wide range of planning and reasoning problems. Notably, it exhibits exceptional proficiency in solving more challenging tasks, like ToolQA, compared to baselines. Additional results in Appendix E and F show that ToolChain* can generalize to a wide range of complex reasoning tasks and open-source LLMs (*e.g.*, LLaMA 2 (Touvron et al., 2023)). (5) There is a trade-off between search depth (*i.e.*, limitations on the number of steps) and the quality of the solution path (Figure 6). ToolChain* efficiently searches optimal solutions within limited steps, striking a **balance between exploration and exploitation**.

## 5 RELATED WORKS

**LLMs for Tool Use.** Recent advances have leveraged LLMs as autonomous agents to master tools and generate solution plans for complicated problems (Qin et al., 2023a;b; Mialon et al., 2023; Shi et al., 2024). Interacting with various tools, LLM agents can augment themselves with real-time factual knowledge (Nakano et al., 2022; Yang et al., 2023a), multi-modality understanding (Shen et al., 2023; Lu et al., 2023; Yang et al., 2023c), computational abilities (Schick et al., 2023; Parisi et al., 2022), code interpretabilities (Gao et al., 2022; Paranjape et al., 2023), and domain-specific functionalities (Zhang, 2023; Jin et al., 2023). However, many existing methods either concentrate on individual tool-use scenarios (Schick et al., 2023; Parisi et al., 2022) or simply inject human-made heuristic ordering rules for multi-tool utilization (Shen et al., 2023; Lu et al., 2023). With the increasing number of potential API functions at each step and the escalating sequence of actions for complex problem solutions, the action space expands exponentially, thereby diminishing their effectiveness. ToolChain* frames the planning challenge across various tools as navigation through the action space to efficiently identify a valid solution path.

**LLMs with Search Algorithms.** The majority of LLM-based agents with open- or closed-loop systems rely on linear reasoning or planning structure. To explore multiple branches in the action space, self-consistency (Wang et al., 2022b) samples multiple chains of thoughts, which can be considered as multiple i.i.d. solution paths in the decision space, selecting the best answer through majority voting. Maieutic prompting (Jung et al., 2022) generates a tree of explanations, enforcing logical consistency. Xie et al. (2023) adopts beam search to decode and improve Chain-of-Thoughts reasoning chain. CoRe (Zhu et al., 2023) proposes to fine-tune both the reasoning step generator and verifier to solve math word problems, incorporating MCTS for reasoning decoding. Tree-of-Thoughts (Yao et al., 2023a) utilizes heuristic approaches, including depth- and breadth-first search to identify better reasoning pathways. Additionally, RAP (Hao et al., 2023a) combines a world model with rewards within an advanced MCTS search approach. However, many search-guided planning approaches face the trade-off between efficient exploration of an expansive action space against the effective exploitation of global optimal solutions. To avoid exhaustive exploration like MCTS, we propose ToolChain* to combine efficient A* search with the effective reasoning ability of LLMs.

## 6 CONCLUSION

In this paper, we propose ToolChain*, an A* tree search-based planning algorithm to augment LLMs with external tools for complicated real-world planning and reasoning tasks. Compared to existing open- or closed-loop LLM agents, ToolChain* formulates the action space as a decision tree, thereby effectively mitigating error propagation and extensively expanding the search space. Furthermore, ToolChain* significantly accelerates the search process compared to other tree search-based methods, enabling tree search in complicated action space and striking a dynamic balance between exploration and exploitation. By achieving significant improvements over state-of-the-art baselines, ToolChain* showcases its potential as an efficient planning algorithm, navigating LLM-based agents in addressing complex real-world challenges.

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

## A  BROADER IMPACTS AND LIMITATIONS

In this study, we introduce ToolChain*, an efficient tree search-based planning algorithm for LLMs-based agents tackling challenging tasks involving tool usage. As a flexible plug-and-play framework with compositional planning and reasoning capabilities, ToolChain* holds considerable promise for positive social impact across diverse domains, including but not limited to real-world tool utilization and complex decision-making processes. One potential limitation to consider is that, while the efficiency of our proposed ToolChain* surpasses other tree search methods and is even comparable to closed-loop systems, it still cannot match the efficiency of open-loop systems. Given the demands for efficiency in real-world applications, we intend to further refine our tree structure and search strategy in future work.

## B  IMPLEMENTATION DETAILS

### B.1  HARDWARE INFORMATION

All experiments are conducted on CPU: Intel(R) Xeon(R) Platinum 8275CL CPU @ 3.00GHz and GPU: NVIDIA A100 SXM4 80 GB using Python 3.8.13.

## B.2 Parameter Configuration

We chose the GPT-3.5-turbo engine for ChatGPT and the GPT-4 engine for GPT-4 when structuring the LLM-based agent. The maximum length for generated solutions is set to 512, and the temperature is set to 1 for flexibility in self-consistency frequency function $g_{t,1}$ (Section 3.2). For LLaMA-2 experiments, the maximum length for generated solutions is set as 256 and the temperature is set to 1. We use 8 NVIDIA A100 SXM4 80 GB GPUs and FastChat (Zheng et al., 2023) to tune the LLaMA-2 7B and 13B models on the training data discussed in Appendix F. For ToolChain*, we set the weights of geometric means between heuristic and non-heuristic parts in cumulative and future costs as $\alpha = \beta = 0.5$ by default. The number of potential actions in self-consistency frequency is set as $k = 10$ and the maximum step limit is set as $T = 20$ by default. Our code repository is presently undergoing an internal review within the company for public release. Upon receiving approval, we will make both the code and data available on GitHub.

## B.3 Task Setup for ToolBench

We define the nodes in the decision tree as function API calls. From the root node, which corresponds to the input task description, we represent the complex process of identifying a successful solution as navigating a valid path in the tree search space. To expand from the current node, we prompt LLMs with API definitions, demonstration examples, the given query, and action history, thereby generating multiple i.i.d. next steps. In order to precisely evaluate plan quality and gauge the proximity of the current action to the goal, we employ task-specific heuristic functions (Section 3.2) for both cumulative and projected scores. To ensure a fair comparison, all prompts dispatched to the baselines and ToolChain* follow the configuration set by ToolBench (Xu et al., 2023). Detailed descriptions of prompts are available in Appendix I.1. We adopt the success rate as the evaluation metric for Home Search, Trip Booking, and Google Sheets. For Virtual Home, we report the proportion of scripts achieving the correct end state or successfully completing the task.

## B.4 Task Setup for GSM8K

GSM8K (Cobbe et al., 2021) serves as a dataset for high school-level mathematical reasoning. Numerical reasoning tasks within this dataset typically comprise a descriptive component followed by a culminating question. Answering this question requires multi-step mathematical calculations based on the context of the description. Notably, the complexity directly relates to the number of mathematical reasoning steps required for a solution. We conduct experiments on the entire set of GSM8K and also a subset of hard questions from GSM8K collected in ToolQA (Zhuang et al., 2023). In adapting ToolChain*, we represent reasoning steps in a solution as nodes in the decision tree. With the math question description as the root node, complex reasoning across numerous intermediate steps is translated into navigating the decision space for a valid path. Given the absence of seed data for long-term memory, we simplify ToolChain* to leverage self-consistency frequency and LLM imagination as the cumulative score $g(n)$ and future score $h(n)$, respectively.

## C LLM-based Agents

We summarize existing LLM-based Agents for tool-use scenarios in Table 4, with a detailed definition and related works of open- and closed-loop systems in the following sections.

### C.1 Open-Loop System

**Open-loop Systems** (Wei et al., 2022; Zhou et al., 2022) generate pre-defined plans to explore over coherent intermediate steps toward problem-solving. Given the initial states $s_0$, which are usually the problem description $x$ in the format of natural language, the systems generate the entire $T$-step plans as the solution $\rho(a_1, a_2, \cdots, a_T | s_0) : \mathcal{S} \rightarrow \Delta(\mathcal{A}^T)$, where $\mathcal{S}$ is the state space, $\mathcal{A}$ is the action space, and $\Delta(\cdot)$ is probability simplex function. The final output is obtained after executing the entire plan $y \sim p(y | s_0, a_1, a_2, \cdots, a_T)$.

**LLMs for Open-Loop Planning.** Trained on vast world knowledge and human examples, LLMs have emerged with smart planning and decision-making capabilities. Leveraging LLMs as autonomous

Table 4: A comparison of methods that leverage LLMs for tool-use. Each method's features are reported into five categories: 1) Basic Information: The basic information of the method about tool-use relevance, covering the total number and type of tools used by the methods, the number of tools involved in solving each problem, the instruction type of methods, and the tasks for evaluation. 2) Planning Ability: The method will generate a complete plan to solve problems. 3) Modify Steps: The method makes changes to the next action (Taking Action). 4) Modify Plans: The method might revise the entire plan (Modifying Plan) and make changes accordingly or not. 5) Trace Back: The method can revise the past actions in the plan and make modifications to restart from previous actions.

| Methods | #Tools | Tool Types | #Tool/Task | Instruction Type | Task | Planning | Modify Steps | Modify Plans | Trace-Back |
|---|---|---|---|---|---|---|---|---|---|
| *Open Loop Methods* | | | | | | | | | |
| CoT (Wei et al., 2022) | 1 | - | 1 | Prompting | QA | ✔ | ✘ | ✘ | ✘ |
| Lila (Mishra et al., 2022) | 1 | Code (Math) | 1 | Prompting | MathQA | ✔ | ✘ | ✘ | ✘ |
| PoT (Chen et al., 2022) | 1 | Code (Math) | 1 | Prompting | TabQA | ✔ | ✘ | ✘ | ✘ |
| Code4Struct (Wang et al., 2022a) | 1 | Code (Event) | 1 | Prompting | Extraction | ✔ | ✘ | ✘ | ✘ |
| PAL (Gao et al., 2022) | 1 | Code (Math) | 1 | Prompting | MathQA | ✔ | ✘ | ✘ | ✘ |
| MathPrompt (Imani et al., 2023) | 1 | Code (Math) | 1 | Prompting | MathQA | ✔ | ✘ | ✘ | ✘ |
| ToolFormer (Schick et al., 2023) | 5 | Basic | 1 | PR & FT | QA | ✔ | ✘ | ✘ | ✘ |
| GraphToolFormer (Zhang, 2023) | 5 | Graph | 1 | PR & FT | Graph | ✔ | ✘ | ✘ | ✘ |
| Talm (Parisi et al., 2022) | - | Basic | 1 | PR & FT | QA | ✔ | ✘ | ✘ | ✘ |
| HuggingGPT (Shen et al., 2023) | >10 | Vision Models | >1 | Prompting | VQA | ✔ | ✘ | ✘ | ✘ |
| Chameleon (Lu et al., 2023) | >10 | NLP, Table | >1 | Prompting | QA | ✔ | ✘ | ✘ | ✘ |
| GeneGPT (Jin et al., 2023) | 38 | NCBI APIs | >1 | Prompting | Gene Tasks | ✔ | ✘ | ✘ | ✘ |
| *Greedy Closed-Loop Methods* | | | | | | | | | |
| WebGPT (Nakano et al., 2021) | 10 | Web Operations | >1 | Fine-tuning | QA | ✔ | ✔ | ✘ | ✘ |
| ART (Paranjape et al., 2023) | 8 | Code, Basic | >1 | Prompting | BigBench | ✔ | ✔ | ✘ | ✘ |
| ReAct (Yao et al., 2023b) | 3 | Retriever | >1 | PR & FT | QA, Env | ✔ | ✔ | ✘ | ✘ |
| MM-ReAct (Yang et al., 2023c) | >10 | Vision Models | >1 | Prompting | CV tasks | ✔ | ✔ | ✘ | ✘ |
| Visual ChatGPT (Wu et al., 2023) | >10 | Vision Models | >1 | Prompting | CV tasks | ✔ | ✔ | ✘ | ✘ |
| *Closed-Loop Methods* | | | | | | | | | |
| DEPS (Wang et al., 2023) | - | Game | - | PR & FT | Game | ✔ | ✔ | ✔ | ✘ |
| AdaPlanner (Sun et al., 2023) | - | Actions | - | Prompting | Envs | ✔ | ✔ | ✔ | ✘ |
| *Tree Search-based Methods* | | | | | | | | | |
| ToT (Yao et al., 2023a) | - | Thoughts | - | PR | Planning | ✔ | ✔ | ✔ | ✔ |
| MCTS (Hao et al., 2023a) | - | Thoughts | - | PR | Planning | ✔ | ✔ | ✔ | ✔ |
| **ToolChain**[*] | >10 | General | >1 | PR & FT | Envs, MathQA | ✔ | ✔ | ✔ | ✔ |

agents to accomplish decision-making tasks has gained attention and shown potential. Earlier studies apply the open-loop framework to directly generate the entire plan as the solution. One line of works, including Chain-of-Thought (Wei et al., 2022) and Zero-Shot Planner (Huang et al., 2022a), generate intermediate reasoning steps all at once to solve the problem. Another line of works selects the opposite direction (*e.g.*, least-to-most prompting (Zhou et al., 2022)) that decomposes the complicated problems into relatively easier sub-problems. For more complex tasks, methods like HuggingGPT (Shen et al., 2023) and Chameleon (Lu et al., 2023) incorporate a set of functional tools and directly generate the plan of API calls in a sequence for execution. However, all these aforementioned methods explore a predetermined single path in the decision space, leaving the rest potential plans not considered when solving the problems.

## C.2 Closed-Loop System

**Closed-loop Systems** follow a more step-by-step plan modification and execution manner, interleaving intermediate observations with decisions over the action space. When the agent interacts with the environment at the $t$-th step, the agent is aware of the current observation $o_t \in \mathcal{O}$ from the environment and generates a trajectory-like context $c_t = (s_0, a_1, o_1, a_2, \cdots, o_t)$. In tool-use scenarios, the intermediate observations are obtained during the execution of the previous actions in the plan $o_t \sim \pi(o_t|s_0, a_1, o_1, \cdots, a_{t-1})$. According to the environment feedback $o_t$, two levels of refinements can be applied: greedy closed-loop methods (Yao et al., 2023b; Shinn et al., 2023) only decide the next single step $a_t \sim \rho(a_t|s_0, c_t; \mathcal{I}, \mathcal{D})$, while the standard ones (Wang et al., 2023; Sun et al., 2023) will modify the entire future plans $\pi(a_{t+1}, a_{t+2}, \cdots, a_{T_g}|s_0, c_t; \mathcal{I}, \mathcal{D})$.

Table 5: Task overview of (1) Home Search, (2) Trip Booking, (3) Google Sheets, (4) Virtual Home, and (5) GSM8K datasets. We provide examples of task descriptions with output actions, and report the number of APIs (# APIs) and questions (# Ques) within each dataset.

| Datasets | Task Descriptions | Output Actions | # APIs | # Ques | # APIs/Sol |
|---|---|---|---|---|---|
| **Home Search** | I want to buy a townhouse, mobile or co-op in Pittsburgh with 4 rooms. My budget is $1385000. | API.set_location("Pittsburgh") API.set_buy_or_rent("buy")··· | 15 | 100 | 7.5 |
| **Trip Booking** | Could you help me find train tickets for 3 children and 5 adults from Des Moines to Cape Coral on July 07, 2022? | API.select_booking_type("trip tickets") API.select_transportation("train")··· | 20 | 120 | 13.4 |
| **Google Sheets** | Update chicken's price by 2. [A table is flattened in the context.] | df = get_as_dataframe(worksheet) df.loc[df['Product'] == 'chicken', 'Price'] += 2··· | 108 | 70 | 6.1 |
| **Virtual Home** | Read book | Agent.Find(novel) Agent.TurnTo(novel)··· | 40 | 100 | 13.4 |
| **GSM8K (ToolQA)** | Elsa has 5 apples. Anna has 2 more apples than Elsa. How many apples do they have together? | 1. Anna has 2 more apples than Elsa. 2. So Anna has 2 + 5 = 7 apples.··· | - | 1319 (100) | - |

**LLMs for Closed-Loop Planning.** Inspired by traditional reinforcement learning approaches that heavily rely on interaction with the environment (either simulator or real world), researchers start to improve the plan via refinement according to the environment feedback. Initially, ReAct (Yao et al., 2023b) and Inner Monologue (Huang et al., 2022b) allow LLM agents to refine a single step greedily according to the environment feedback. Considering that solely modifying the immediate action being executed is easy to fall into local sub-optimal actions, improvements like DEPS (Wang et al., 2023) and AdaPlanner (Sun et al., 2023) are proposed to recursively modify the entire future plans according to the environment feedback. However, without a tracing-back mechanism to check the already executed plans, these efforts in the closed-loop framework still only explore a small proportion of decision space. To mitigate these issues, we propose ToolChain*, that enables tree search in planning and can record multiple branches in the decision space.

## D EXPERIMENTAL DETAILS

### D.1 DATASET DETAILS

We evaluate ToolChain* on four tool-use environments in ToolBench (Xu et al., 2023) and one reasoning task in GSM8K (Cobbe et al., 2021). Table 5 provides examples of task descriptions and output actions and reports the number of APIs and questions incorporated in the environment.

• **Home Search** simulates the process of locating homes in a specific area based on certain criteria. Agents are required to leverage 15 functions (*e.g.*, set_location, search, *etc.*), to aid users in completing the search.

• **Trip Booking** is a task that parallels the Home Search but incorporates more advanced dependencies between function calls. This task simulates the process of submitting search requests for transportation tickets, hotel rooms, or a combination of both. Specifically, it is guided by specific searching conditions or parameters like locations, dates, and the number of tickets. The API includes a total of 20 functions related to trip booking scenarios.

• **Google Sheets** involves manipulating actual worksheets from Google Sheets [1] via the gspread library [2], including common operations such as updating cell values, sorting, *etc.*.

• **Virtual Home** derives from the setting of the VirtualHome simulator (Puig et al., 2018). It requires the LLM agents to generate sequences of actions with multiple function calls for completing household activities. It consists of 40 functions (*e.g.*, Sleep(), Push(object), and PourInto(object1, object2)), each corresponding to a specific action exemplified in the

---

[1] https://www.google.com/sheets/about/
[2] https://docs.gspread.org/en/latest/

simulator. Every function can take up to two arguments, representing valid objects within the household settings.

• **GSM8K** (Cobbe et al., 2021) is a dataset of high-quality linguistically diverse grade school math word problems. To evaluate the model performance on the GSM8K dataset, we evaluate the accuracy at which models are able to obtain the final correct answer. In addition, we also report model performance on a subset of hard questions in GSM8K, curated by **ToolQA** (Zhuang et al., 2023). The hard questions are sampled from the error cases made by ChatGPT on the original GSM8K dataset.

## D.2 Baseline Details

For tool-use tasks in ToolBench, we compare ToolChain* with GPT (OpenAI, 2023), ReAct (Yao et al., 2023b), AdaPlanner (Sun et al., 2023), Tree-of-Thoughts (Yao et al., 2023a), and MCTS (Hao et al., 2023a). For math reasoning tasks in GSM8K, we compare ToolChain* with the state-of-the-art reasoning approaches, including GPT (OpenAI, 2023), Chain-of-Thought (Wei et al., 2022), Self-Consistency (Wang et al., 2022b), ReAct (Yao et al., 2023b), Tree-of-Thoughts (Yao et al., 2023a), and MCTS (Hao et al., 2023a).

We provide details of each baseline as follows:

• **GPT** (OpenAI, 2023) is a standard input-output prompt with four in-context examples. We directly send the four-shot examples with the task description together to GPT-3.5-turbo and GPT-4. GPT series serve as open-loop systems with closed-source LLMs as backbones.

• **Chain-of-Thought** (Wei et al., 2022) prompting is an open-loop mechanism to enhance reasoning in language models. Chain-of-thought prompting breaks down multi-step problems, allocating additional computational resources for extensive reasoning stages step by step.

• **Self-Consistency** (Wang et al., 2022b) is an open-loop decoding strategy to replace the conventional greedy decoding typically employed in chain-of-thought prompting. It initially samples an array of diverse reasoning paths rather than solely relying on the most probable one. Subsequently, by marginalizing over these sampled reasoning trajectories, it pinpoints the most coherent answer.

• **ReAct** (Yao et al., 2023b) integrates reasoning with tool use by prompting LLMs to generate interleaved verbal reasoning traces and tool calls. ReAct is a closed-loop system for LLM-based agents in tool usage.

• **AdaPlanner** (Sun et al., 2023) is a closed-loop system that enables the LLM agent to adaptively refine its self-conceived plan based on feedback from the environment. This refinement process leverages both in-plan and out-of-plan strategies.

• **Tree-of-Thoughts** (Yao et al., 2023a) is a tree search-based LLM reasoning framework that built upon chain-of-thought prompting. It facilitates exploration across cohesive textual segments, which act as intermediary steps in the problem-solving process. Unlike linear reasoning pathways in open- or closed-loop systems, ToT enables language models to engage in decision-making by examining various reasoning trajectories. With the ability to self-assess choices for the subsequent steps, ToT provides the flexibility to anticipate future steps or revisit previous ones to make holistic decisions.

• **MCTS** (Hao et al., 2023a) is a tree search-based LLM reasoning framework, repurposing LLM and functioning both as a world model and a reasoning agent. It integrates a principled planning algorithm, specifically based on MCTS, facilitating strategic exploration in the expansive reasoning space. LLM-based agent systematically constructs a reasoning tree, guided by its inherent world model and task-specific rewards.

## D.3 Additional Analysis on ToolBench

We observe that closed-loop methods typically perform better than tree search-based methods on the Home Search and Trip Booking datasets. Conversely, for the Google Sheets and Virtual Home datasets, tree search-based methods perform better. This discrepancy can be attributed to the nature of feedback provided by the datasets. For instance, the Home Search and Trip Booking datasets offer precise environmental feedback regarding plan errors (*e.g.*, "*Task execution error: 'HomeSearchAPI' object has no attribute 'set_min_baths'*"), enabling closed-loop systems to effectively modify their plans. In contrast, the Google Sheets and Virtual Home datasets, with their extensive API function

calls (108 and 40, respectively), present a notably larger action space than Home Search and Trip Booking (15 and 20, respectively).

## D.4 CASE STUDIES

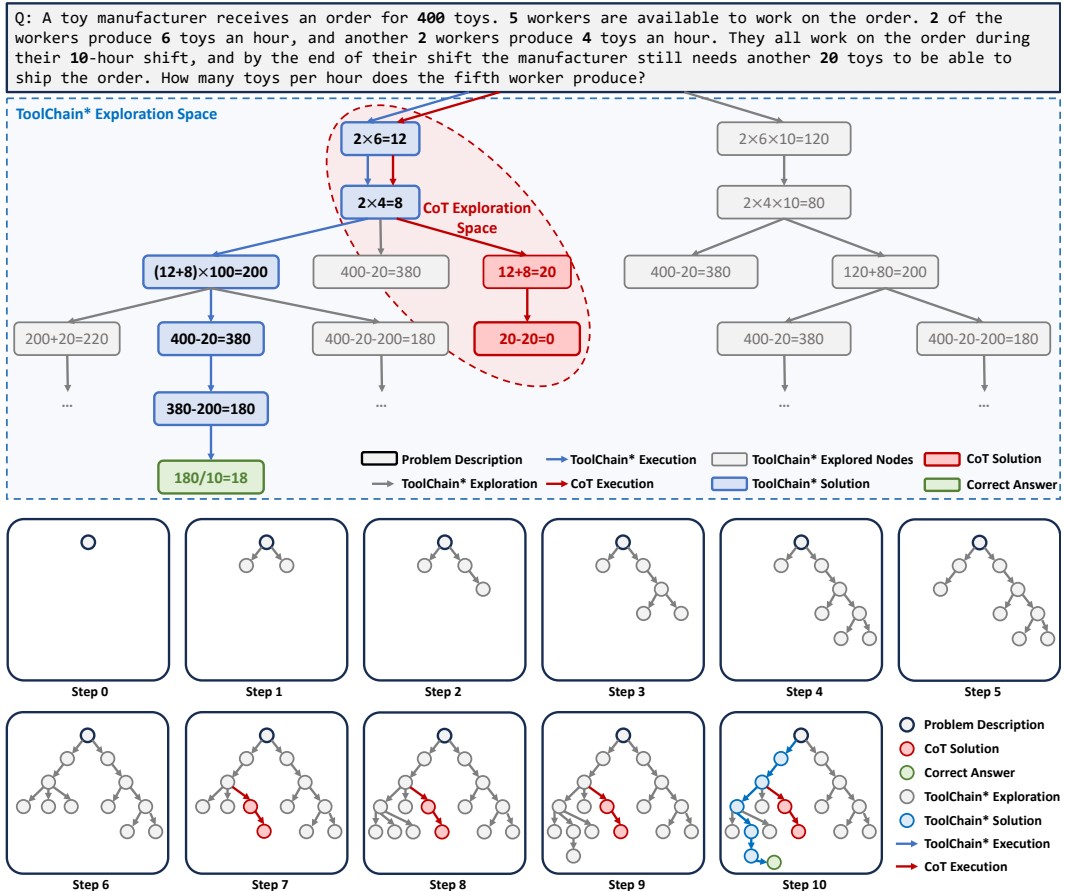

Figure 7: Case study comparing Chain-of-Thought and ToolChain* on GSM8K dataset. We offer comparison in exploration space (upper part) and planning ordering (lower part). Given the input query, ToolChain* explores a wide range of potential nodes (blue shadowed area) in the decision tree, while Chain-of-Thought only explores one direction (red shadowed area). During the planning process, a chain of thought can fall into a faulty loop or a dead end with a previous incorrect action. However, ToolChain* can gradually abandon the faulty path by increasing the cost after the incorrect action. This enables it to revise previous actions and jump out of the faulty path to try another plan.

We include additional case studies with visualizations comparing the Chain-of-Thought and ToolChain* on the GSM8K dataset. As shown in Figures 7 and 8, we compare the exploration space (upper part) and planning ordering (lower part) of both methods on the same problem. Given the input query, ToolChain* explores a wide range of potential nodes (blue) in the decision tree, while Chain-of-Thought only explores one direction (red). Moreover, from a step-by-step visualization of the planning process in Figure 7, we notice that chain-of-thought falls into a faulty loop or a dead end with a previous incorrect action. However, ToolChain* can gradually discard the faulty path by increasing the cost after the incorrect action. This allows our method to re-evaluate and adjust prior actions, facilitating a shift away from erroneous paths to explore alternative plans. More importantly, it mitigates the error propagation along the action plan, which usually occurs in linear or unidirectional solutions (*i.e.*, open- and closed-loop systems).

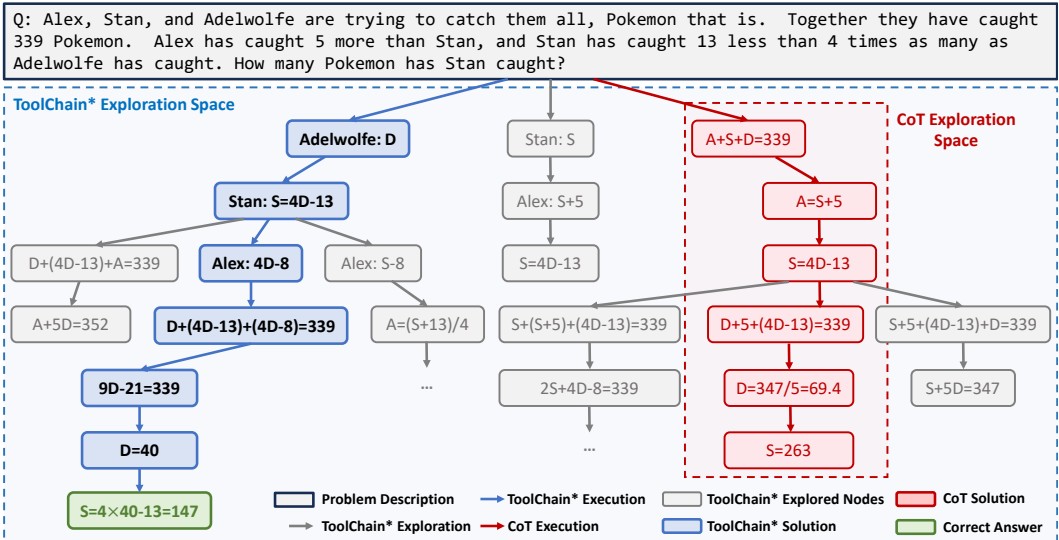

Figure 8: Case study comparison between Chain-of-Thought and ToolChain* on the GSM8K dataset. The exploration space is illustrated in the upper section, and the planning order is depicted in the lower section. For a given input query, ToolChain* explores an expansive set of potential nodes (blue) with correct answers, whereas Chain-of-Thought primarily navigates in a singular direction (red).

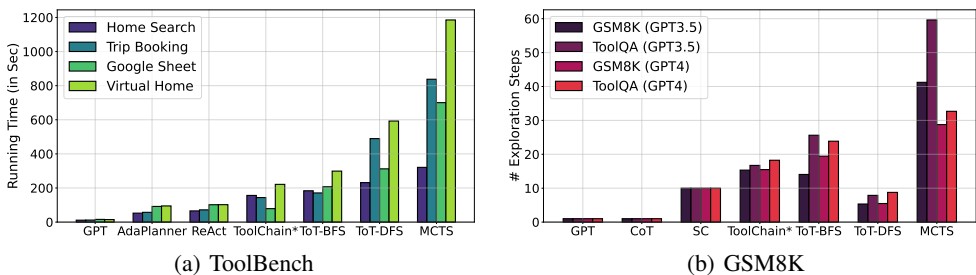

(a) ToolBench

(b) GSM8K

Figure 9: Additional time efficiency evaluation on ToolBench and GSM8K. (a) We report additional average running time in seconds over all instances in the dataset using GPT-4 backbone. (b) We calculate the average number of valid actions in GSM8K for math reasoning efficiency analysis. In both cases, ToolChain* shows close efficiency to closed-loop systems without a tree structure and outperforms other tree-search algorithms in terms of efficiency.

## D.5 EFFICIENCY DETAILS

In terms of additional efficiency analysis in tool use, we evaluate the running time of ToolChain* against all the baselines based on GPT-4, as shown in Figure 9(a). Consistent with our previous results, ToolChain* is faster than the most efficient tree search-based method, Tree-of-Thoughts (BFS). For the math reasoning task, we conduct efficiency analysis with a number of valid actions in Figure 9(b). We calculate the average number of valid actions in GSM8K for math reasoning efficiency analysis. Similarly, ToolChain* shows close efficiency to closed-loop systems without a tree structure and outperforms other tree-search algorithms in terms of efficiency.

From the scaling-up analysis in Figure 10, we can explicitly identify a crucial trade-off between effectiveness and efficiency in the direct application of tree search-based reasoning methods to complex tool use scenarios, including ToT-DFS (Yao et al., 2023a), ToT-BFS (Yao et al., 2023a), and MCTS (Hao et al., 2023a). Compared with ToolChain*, which quickly converges on with time efficiency and high-quality solution, the rest tree-based search methods not only suffer from relatively low success rate (Figure 10(a)), but also struggle with the long running time (Figure 10(b)). This could be further verified by experimental results in Table 1 and efficiency metrics in Figure 5.

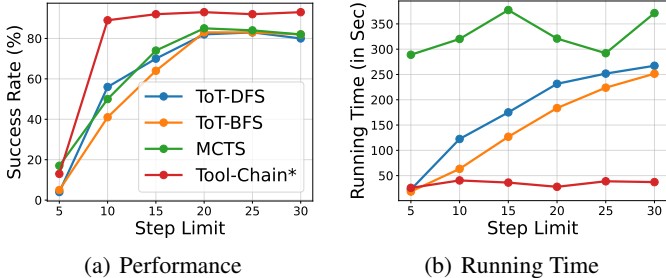

|                    |                   |
|:------------------:|:-----------------:|
| (a) Performance    | (b) Running Time  |

Figure 10: Scaling analysis of ToT-DFS (Yao et al., 2023a), ToT-BFS (Yao et al., 2023a), MCTS (Hao et al., 2023a), and ToolChain*. (a) Performance and (b) running time on Home Search when scaling up step limitations $T$.

### D.6 ABLATION STUDIES DETAILS

We conduct a detailed analysis of the ablation studies for both the cumulative and future cost functions in the following, as presented in Table 2. First, experimental results indicate that each element within both the cumulative and future cost functions enhances the performance of ToolChain*. This verifies the efficacy of our proposed cost functions in guiding the search through the decision tree. Second, dropping either the heuristic components ($g_{1,t}$, $h_{1,t}$) or the non-heuristic ($g_{2,t}$, $h_{2,t}$) components from the cumulative or future cost functions results in a decline in performance. Specifically, when the non-heuristic components are absent, there is an average drop in the success rate of $6.8\%$; whereas, without the heuristic components, the decrease is $3.3\%$. This suggests that in most instances, non-heuristic components have a more significant impact. This is potentially because heuristic components, relying on long-term memory, are limited in covering all test cases and may struggle to precisely estimate the cost of new actions or tasks. In scenarios where environments offer high-quality seed data for long-term memory (*e.g.*, Virtual Home), the heuristic function plays a more important role. Third, eliminating either the entire cumulative or future cost results in a marked decline in the success rate. Relying exclusively on the future cost induces a sharp performance drop of $28.2\%$, deteriorating ToolChain* to a greedy strategy that favors the shortest solution plans with the least number of actions. Conversely, if the search is guided only by the cumulative cost, ToolChain* essentially mirrors the behavior of the BFS algorithm, yielding similar performance outcomes.

## E  ADDITIONAL DATASET: SCIENCE QA

Table 6: Additional results on ScienceQA based on GPT-3.5-turbo.

| Models                       | Accuracy |
|------------------------------|:--------:|
| CoT (GPT-3.5-turbo)          | 78.3     |
| Chameleon (GPT-3.5-turbo)    | 79.9     |
| ToolChain* (GPT-3.5-turbo)   | **80.7** |

We evaluate ToolChain* in Science Question Answering (ScienceQA) (Lu et al., 2022) dataset on general reasoning tasks. ScienceQA serves as a diverse benchmark for multi-modal question answering across an array of scientific topics and contexts. Answering its questions requires a variety of tools and capabilities, including image captioning, text detection, knowledge retrieval, online resource searches, and multi-clue visual reasoning. Since ToolChain* can function as a plug-and-play planner, we replace the original naive LLM-based planner in Chameleon with ToolChain* for evaluation purposes. For our experiments, we select the Chain-of-Thought (Wei et al., 2022) and the original Chameleon (Lu et al., 2023) as reference baselines. When built upon GPT-3.5-turbo as the foundational LLM, ToolChain* realizes an accuracy improvement of $0.8\%$ over Chameleon. ToolChain* benefits from considering multiple potential paths for execution. However, the performance gain is limited on the dataset, primarily because the logic behind calling different tools to solve the problem from the dataset is quite rigid.

## F  OPEN-SOURCE LLMS

Table 7: Additional experimental results (success rate) on Virtual Home for tool-use evaluation based on LLaMA-2 (Touvron et al., 2023) 7B and 13B models.

| Models | Executable | Success | Success (E) |
|--------|-----------|---------|-------------|
| LLaMA-2 (7B) | 31.0 | 21.2 | - |
| LLaMA-2 (13B) | 30.0 | 19.8 | - |
| StepLLaMA-2 (7B) | 57.0 | 25.4 | 29.5 |
| StepLLaMA-2 (13B) | 56.0 | 25.3 | 27.4 |
| ToolChain* (Step-LLaMA-2, 7B) | 57.0 | 28.7 | **31.0** |
| ToolChain* (Step-LLaMA-2, 13B) | **96.0** | **30.0** | 30.2 |

**Task Setup** In this section, we explore the potential of leveraging ToolChain* on open-source large language models. ToolBench (Xu et al., 2023) offers the training data for each of the datasets. The instructions in the original training data include the task query $g$ and a simple "`Action:`" prompt to make the model generate the entire plan $p = (a_0, a_1, \cdots, a_{T_g})$. However, without any API documentation or demonstration examples provided in the instructions, generating a comprehensive plan proves challenging for models. Additionally, we strive to further refine the connections between adjacent actions in the solution plans within the training data. Thus, we decompose the solution plans autoregressively, assembling the original task query with the previous $t$ actions as instruction and treating the $(t + 1)$-th action in the plan as the solution. More specifically, for step $t$ and the task query $g$, the new instructions are in the format of $(g, a_0, a_1, \cdots, a_t)$. The corresponding solution is the next-step action $a_{t+1}$. In addition, with the decomposition, the size of training data increases from $\sim 500$ to $\sim 4500$. With the generated instruction-solution pairs, we fine-tune two StepLLaMA-2 models based on LLaMA-2 (Touvron et al., 2023) 7B and 13B models. As ToolChain* can be applied as a plug-and-play module for agents based on different LLMs, We also equip the fine-tuned StepLLaMA-2 with ToolChain*.

**Results** We evaluate the performance of both StepLLaMA-2 and ToolChain* (StepLLaMA-2) on the Virtual Home dataset and show the results in Table 7. We report the metrics of Executable, Success, and Success (E). Executable means the proportion of plans that can be executed in VirtualHome without violating any rules. Success means the proportion of plans that lead to the correct final state. Success (E) is a variant of Success, which only tests the Success rate on executable plans. StepLLaMA-2 and ToolChain* (LLaMA-2) both outperform the LLaMA-2 models that are directly tuned on the original training data in ToolBench. In addition, applying ToolChain* on StepLLaMA-2 can bring $4.0\%$ improvement in success rate on average, showing that ToolChain* can also be effective on open-source LLMs.

## G  ADDITIONAL ANALYSIS ON VALUE FUNCTIONS

Table 8: Additional results on ToolChain* with different value functions.

| | Home Search | Trip Booking | Google Sheets | Virtual Home | Average |
|--|-----------|------------|-------------|------------|---------|
| ToolChain* (Self-Evaluation Hao et al. (2023a)) | 84.0 | 83.3 | 49.9 | 21.5 | 59.7 |
| ToolChain* (ToT-Vote Yao et al. (2023a)) | 82.0 | 81.7 | 53.4 | 21.0 | 59.6 |
| **ToolChain*** | **93.0** | **90.8** | **61.4** | **28.6** | **68.5** |

To validate the advantage of our proposed value function, we explored various cost function formulations, including self-evaluation prompting LLMs to generate verbalized scores (Hao et al., 2023a), and vote scores in 'tree-of-thoughts' (Yao et al., 2023a). Our findings indicated that these scoring methods were not consistently accurate. We identified two key factors guiding our heuristic function design: (1) the black-box nature of state-of-the-art LLMs offering limited information for score formulation, and (2) the observation that similar tasks often share similar logical structures in their solution plans. Driven by the two motivations, we developed heuristic scores to maximize

the use of additional information derived from past experiences (Table 8). The results demonstrate that a combination of LLM-generated scores and heuristic scores yields superior performance. In the initial stages, due to limited data in the long-term memory, the model predominantly relies on the LLM-generated scores for generating plans. As the long-term memory expands, the heuristic functions increase in accuracy, effectively regularizing and refining the LLM-generated scores.

Table 9: Additional results on ToolChain* with different value functions using LLaMa-2 (7B).

|  | Virtual Home |
|---|---|
| ToolChain* (Action Likelihood) | 25.7 |
| ToolChain* (Answer Probability) | 26.1 |
| ToolChain* (Reward Function) | 27.4 |
| **ToolChain*** | **28.7** |

Furthermore, we have conducted explorations to utilize additional information from open-sourced LLMs in order to formulate a more precise cost function. These explorations include: (1) action likelihood, calculated as the product of token log probabilities within an action; (2) answer probability, determined by the token probability of 'yes' in response to the question 'Is this action step correct?'; and (3) a reward-style probability, which applies weight decay based on the length of tokens in the actions and the total number of actions in a plan. Table 9 presents additional experimental results using LLaMA-2 (7B). The results indicate that our proposed cost functions outperform other value functions from open-source LLMs.

## H   COST ANALYSIS

We provide more detailed information regarding the computational costs of ToolChain* (GPT-3.5) across our datasets in Table 10. It is important to note that the heuristic functions in ToolChain* do not require additional LLM calls. Instead, they rely on a long-term memory that stores solution plans. As for the costs related to the environment or domain simulations, these are primarily included in the prompts for expansion. This aspect accounts for approximately $7.22. Similar budgetary considerations apply to recent advances (Yao et al., 2023b; Sun et al., 2023; Yao et al., 2023a) that also rely on experiments utilizing OpenAI APIs. Additionally, it is worth noting that the cost of using gpt-3.5-turbo is progressively decreasing within months, making it more affordable in the future.

Table 10: Additional computational costs of ToolChain* based on gpt-3.5-turbo.

| Costs | Expansion | Heuristic Functions | LLM-based Functions |
|---|---|---|---|
| # LLM Calls per Question | 16.44 | 0 | 107.31 |
| $ Prompt Cost per Dataset | 7.22 | 0 | 5.53 |
| $ Completion Cost per Dataset | 8.23 | 0 | 9.25 |

## I   PROMPTS

### I.1   TOOL USE: TOOLBENCH

We follow the prompt format from ToolBench (Xu et al., 2023), which consists of API documents, three-shot in-context demonstration examples, and the query. We utilize the same retriever as the ToolBench implementation [3] to obtain the pertinent API documents and demonstration examples.

```
                    ┌── <ToolChain*_ToolBench> Prompt ──┐
{api_docs}
{examples}
Task: {query}
Action:
```

---

[3] https://github.com/sambanova/toolbench/tree/main

We then provide examples of API documents and demonstrations for each dataset used in our experiments.

### I.1.1 HOME SEARCH

We present five examples of API documents from the Home Search dataset as follows:

```
────────────────── <ToolChain*_HomeSearch_Doc> Prompt ──────────────────
# To set home types for search. For home buying, home_types choices are:
    "House", "Townhouse", "Condo", "Land", "Multi-family", "Mobile",
    "Co-op"; for home renting, home_types choices are: "House",
    "Townhouse", "Condo", "Apartment".
API.select_home_type(home_types: List[str])

# To specify whether to search homes for buying or renting. 'value' can
    be chosen from ['buy', 'rent']. This function must be called after
    setting the location and before setting any other criteria.
API.set_buy_or_rent(value: str)

# To set the maximum commute time in minite
API.set_max_commute_time(value: int)

# To set the minimum home price in dollars
API.set_min_price(value: int)

# To set the maximum home price in dollars
API.set_max_price(value: int)
```

We also provide one demonstration example from the Home Search dataset:

```
────────────────── <ToolChain*_HomeSearch_Demo> Prompt ──────────────────
Task: I want to buy a townhouse, mobile or co-op in Pittsburgh with 4
    rooms. My budget is $1385000.
Actions:
API.set_location("Pittsburgh")
API.set_buy_or_rent("buy")
API.select_home_type(["Townhouse", "Mobile", "Co-op"])
API.set_num_beds(4)
API.set_max_price(1385000)
API.search()
```

### I.1.2 TRIP BOOKING

We present five examples of API documents from the Trip Booking dataset below:

```
────────────────── <ToolChain*_TripBooking_Doc> Prompt ──────────────────
# To select the transportation type from ['flight', 'train', 'bus',
    'cruise'].
API.select_transportation(transportation_type)

# To select the booking type from ['hotels', 'trip tickets', 'both'].
API.select_booking_type(booking_type)

# To set the number of child tickets to purchase.
API.set_num_children(value)

# To set the number of adult tickets to purchase.
API.set_num_adults(value)

# To set the location for arrival, given a Loc object.
API.set_destination(Loc)
```

We also provide one demonstration example from the Trip Booking dataset:

```
┌──────────────── <ToolChain*_TripBooking_Demo> Prompt ────────────────┐
Could you help me find train tickets for 3 children and 5 adults from
    Des Moines to Cape Coral on July 07, 2022? My budget is up to 280
    per ticket.
Actions:
API.select_booking_type("trip tickets")
API.select_transportation("train")
API.set_num_children(3)
API.set_num_adults(5)
location_from = Loc("Des Moines")
API.set_origin(location_from)
location_to = Loc("Cape Coral")
API.set_destination(location_to)
departure_date = Date(7, 7, 2022)
API.set_departure_date(departure_date)
API.set_max_ticket_price(280)
API.search()
```

### I.1.3 GOOGLE SHEETS

We present four examples of API documents from the Google Sheets dataset as follows:

```
┌──────────────── <ToolChain*_GoogleSheets_Doc> Prompt ────────────────┐
# Sets values in a cell range of the sheet.
worksheet.update(range_name, values=None, **kwargs)

# Updates the value of a cell.
worksheet.update_cell(row, col, value)

# Deletes multiple columns from the worksheet at the specified index.
worksheet.delete_columns(start_index, end_index=None)

# Deletes multiple rows from the worksheet at the specified index.
worksheet.delete_rows(start_index, end_index=None)
```

We also provide one demonstration example from the Google Sheets dataset:

```
┌──────────────── <ToolChain*_GoogleSheets_Demo> Prompt ────────────────┐
| Product | Cost | Price |
| beef | 1 | 3 |
| pork | 5 | 4 |
| chicken | 10 | 11 |
| lamb | 3 | 15 |
| duck | 12 | 2 |
| fish | 2 | 100 |

Task: Sets 'Hello world' in 'A2' cell
Actions:
worksheet.update('A2', 'Hello world')

Task: Sets 'Hello world' in 'A2' cell
Actions:
worksheet.update_cell(2, 1, 'Hello world')

Task: Updates A2 and A3 with values 42 and 43
Actions:
worksheet.update('A2:A3', [[42], [43]])

Task: Updates D2 with values 3
Actions:
worksheet.update('D2', 3)

Task: Sum A1:A4 and write the result below A4
Actions:
```

```
worksheet.update('A5', '=SUM(A1:A4)', raw=False)

Task: Update chicken's price by 2
Actions:
df = get_as_dataframe(worksheet)
df.loc[df['Product'] == 'chicken', 'Price'] += 2
worksheet.clear()
set_with_dataframe(worksheet, df, include_index=False,
    include_column_header=True)
```

### I.1.4 VIRTUAL HOME

Below, we present five examples of API documents from the Virtual Home dataset:

─────── `<ToolChain*_VirtualHome_Doc>` Prompt ───────
```
# Take a piece of clothes off. 'object' can only be: ['clothes_jacket',
    'clothes_dress', 'clothes_hat', 'shoes', 'clothes_shirt',
    'clothes_pants'].
Agent.TakeOff(object)

# Scrub an object. 'object' can only be: ['mop', 'cup', 'toilet',
    'plate', 'soap', 'sink', 'spoon', 'cat', 'shower', 'dishwasher',
    'hands_both', 'drinking_glass', 'bowl', 'towel'].
Agent.Scrub(object)

# Rinse an object. 'object' can only be: ['cup', 'pot', 'water',
    'water_glass', 'sponge', 'soap', 'towel', 'dish_soap', 'oven',
    'cleaning_solution', 'knife', 'spoon', 'sink', 'faucet',
    'clothes_underwear', 'detergent', 'drinking_glass', 'hands_both',
    'toilet', 'shower', 'rag', 'plate', 'bowl', 'fork'].
Agent.Rinse(object)

# Wash an object. 'object' can only be: ['face', 'cup', 'food_vegetable',
    'dresser', 'fork', 'shoes', 'child', 'coffee_cup', 'bed', 'water',
    'soap', 'duster', 'brush', 'bathtub', 'toy', 'cleaning_bottle',
    'hair', 'sink', 'razor', 'hands_both', 'drinking_glass', 'table',
    'toilet', 'basket_for_clothes', 'shower', 'dishwasher', 'plate',
    'bowl', 'spoon'].
Agent.Wash(object)

# Pull an object. 'object' can only be: ['table', 'mop', 'mouse',
    'chair', 'clothes_pants', 'light_bulb', 'curtain', 'vacuum_cleaner',
    'mat', 'cat', 'food_food', 'drawing', 'shoes', 'centerpiece',
    'sheets', 'pot', 'laptop'].
Agent.Pull(object)
```

We also provide one demonstration example from the Virtual Home dataset:

─────── `<ToolChain*_VirtualHome_Demo>` Prompt ───────
```
Task: Put down bags
Actions:
Agent.WalkTo(dining_room)
Agent.WalkTo(food_food)
Agent.Find(food_food)
Agent.Grab(food_food)
Agent.Find(table)
Agent.Put(food_food, table)
```

### I.2 MATH REASONING: GSM8K

Below are the prompts utilized for the math reasoning dataset, GSM8K (Silver et al., 2016):

```
┌─────────────── <ToolChain*␣GSM8K> Prompt ───────────────┐
Please complete the plans to solve the question. Here is several
    examples:
Q: Four years ago, Kody was only half as old as Mohamed. If Mohamed is
    currently twice 30 years old, how old is Kody?
A: Let's think step-by-step:
1. We were told that Mohamed is currently twice 30 years old, so he is
    currently 30 * 2 = 60 years old.
2. That means that four years ago he must have been 60 - 4 = 56 years
    old.
3. Four years ago, Kody was half as old as Mohamed, so Kody must have
    been 56 / 2 = 28 years old then.
4. Since Kody was 28 years old four years ago, she must now be 28 + 4 =
    32 years old.
5. The answer is 32.

Q: Carla bought 2 bags of mini peanut butter cups on clearance. Each bag
    was $6.00 but was 75% off. How much did she spend on 2 bags of candy?
A: Let's think step-by-step:
1. Each bag was $6.00 but was 75% off. So each bag cost $6.00 *
    (1 - 0.75) = $6.00 * 0.25 = $1.50.
2. Carla bought 2 bags. So she spent $1.50 * 2 = $3.00.
3. The answer is 3.

Q: If Pam is currently twice as young as Rena is, and in 10 years Rena
    will be 5 years older than her, how old is Pam now?
A: Let's think step-by-step:
1. Since Rena will be 5 years older than Pam in 10 years, she must be
    5 years older than Pam now as well.
2. If Pam is currently twice as young as Rena, that means that Rena is
    currently twice as old as Pam is.
3. So if P stands for Pam's age now and R stands for Rena's age now,
    then we know that R = 2 * P.
4. And since Rena is 5 years older than Pam now, we know that R = P + 5.
5. By substitution, we have P + 5 = 2 * P, which means that P = 5.
6. The answer is 5.

Q: Cappuccinos cost $2, iced teas cost $3, cafe lattes cost $1.5 and
    espressos cost $1 each. Sandy orders some drinks for herself and
    some friends. She orders three cappuccinos, two iced teas, two
    cafe lattes, and two espressos. How much change does she receive
    back for a twenty-dollar bill?
A: Let's think step-by-step:
1. Sandy ordered three cappuccinos, which cost $2 each, so she spent
    $2 * 3 = $6 on cappuccinos.
2. She ordered two iced teas, which cost $3 each, so she spent
    $3 * 2 = $6 dollars on ice teas.
3. She ordered two cafe lattes, which cost $1.5 each, so she spent
    $1.5 * 2 = $3 on cafe lattes.
4. She ordered two espressos, which cost $1 each, so she spent
    $1 * 2 = $2 on espressos.
5. So altogether, Sandy spent $6 + $6 + $3 + $2 = $17 on drinks, which
    means that sandy will get $20 - $17 = $3 as change.
6. The answer is 3.
[END OF EXAMPLE]
Q: {question}
A: Let's think step-by-step:
└─────────────────────────────────────────────────────────┘
```

