# OpenReview forum: "ToolChain*: Efficient Action Space Navigation in Large Language Models with A* Search"
_ICLR.cc/2024/Conference — ICLR 2024 poster_

### Official Review · Reviewer_7Kjr · 2023-10-28

**Soundness:** 4 excellent
**Presentation:** 4 excellent
**Contribution:** 4 excellent
**Rating:** 8
**Confidence:** 5

**Summary:**

In this paper, the authors propose a tree-search-based planning framework for Large Language Models (LLMs). Their framework is motivated by A* search in heuristic planning and uses a cost function, which is a sum of (past) accumulated cost and (future) heuristic cost to update their planning tree. The use of simple heuristics also makes it more resource-efficient than concurrent works based on Monte Carlo Tree Search. The authors demonstrate both empirically, and through various ablation studies, how their proposed method generates more feasible plans.

**Strengths:**

* Clarity: The paper is well-written and easy to follow.
* Novelty: The proposed tree-search-based method is novel (see footnote) and uses a simple but effective cost-function design that helps in better decision-making.
* Significance: The experiments adequately justify the superior performance of their proposed method, both empirically and qualitatively. The use of heuristics for LLM planning is appealing to the LM planning & reasoning community and could be a potential research direction for the future.

Footnote: Note the work "SayCanPay: Heuristic Planning with Large Language Models using Learnable Domain Knowledge, Hazra, et al., 2023" which also proposes heuristic planning with LLMs. However, given its recency, I have considered it as a concurrent work, and therefore have not compared them despite their overlapping contributions.

**Weaknesses:**

I did not find any major weaknesses in the paper. However, it would be nice to have some clarifications regarding the proposed heuristic functions (see Questions).

**Questions:**

1. What do you mean by "task-specific" heuristic functions? Are the tasks here based on "g"? If so, do you use different memory buffers for each task, and how many trajectories do you need in the memory buffer before it produces a valid score?

2. It seems to me that the future cost heuristic might mislead the planner since it is likely that the same action may appear at different positions (in the sequence) for different tasks. For instance: "Clean a kitchen table" and "Set a kitchen table" have their action sequences reversed.

---

> ### Author Response · Authors · 2023-11-19
> **Initial Response to Reviewer 7Kjr**
>
> We sincerely appreciate Reviewer 7Kjr for considering our work as novel and significant with empirical and qualitative improvements. Your suggestions and insights are very helpful for further enhancing the submission quality. We will include a comparison of concurrent work [1] and include the following changes in our updated draft. Please find the responses below:
>
> **[Question 1] What do you mean by "task-specific" heuristic functions? Are the tasks here based on "g"? If so, do you use different memory buffers for each task, and how many trajectories do you need in the memory buffer before it produces a valid score?**
>
> Similar to the definition in [2], we use “task-specific” to describe the heuristic functions for two main reasons: (1) different seed data are utilized for the long-term memory in each task (i.e., dataset); (2) the computed heuristic values are only meaningful within the context of their respective tasks, as similar questions often share underlying solution logic. All tasks are defined based on the natural language descriptions $g$, with further details in Section 2. The initial number of trajectories in the memory buffers for each dataset is as follows:
> |            | Home Search | Trip Booking | Google Sheets | Virtual Home |
> |------------|:-----------:|:------------:|:-------------:|:------------:|
> | # Initial Plans in Long-Term Memory | 11 | 11 | 10 | 50 |
>
> All seed data for these memory buffers are derived from ToolBench [3], which is a reasonable number of trajectories and comparable to few-shot demonstrations.
>
> **[Question 2] It seems to me that the future cost heuristic might mislead the planner since it is likely that the same action may appear at different positions (in the sequence) for different tasks. For instance: "Clean a kitchen table" and "Set a kitchen table" have their action sequences reversed.**
>
> We acknowledge that the future cost heuristic function may not always be accurate. With experimental results presented in Table 2, we have proposed two solutions to mitigate this issue: (1) we integrated the LLM imagination score with heuristic functions to mutually regularize their outputs; (2) the calculation of the heuristic future function involved averaging the relative positions of each action across different plans. Thus, if there exist two scenarios of “Clean a kitchen table → Set a kitchen table” and “Set a kitchen table → clean a kitchen table”, our method effectively incorporates information from both sequences into the heuristic future score.
>
> **[Reference]**
>
> [1] Hazra et al. “SayCanPay: Heuristic Planning with Large Language Models using Learnable Domain Knowledge.” arXiv 2023.
>
> [2] Hao et al. “Reasoning with language model is planning with world model.” EMNLP 2023.
>
> [3] Xu et al. “On the Tool Manipulation Capability of Open-source Large Language Models.” arXiv 2023.

---

> > ### Comment · Reviewer_7Kjr · 2023-11-20
> > **Thank you authors**
> >
> > Thank you for taking the time to address my doubts. I'm satisfied with the answers and have no further questions for now. Cheers!

---

> > > ### Author Response · Authors · 2023-11-20
> > > **Thank you Reviewer 7Kjr**
> > >
> > > Thank you very much for taking the time to review our rebuttal and offering insightful feedback. We will update our paper with the additional results and discussions. Cheers!

---

### Official Review · Reviewer_tHXf · 2023-10-31

**Soundness:** 3 good
**Presentation:** 4 excellent
**Contribution:** 2 fair
**Rating:** 6
**Confidence:** 4

**Summary:**

Leveraging powerful LLMs in complex decision-making tasks is a promising direction, and the authors discussed their categories with clear explanations. For efficient reasoning, the authors proposed a new method inspired by A* search and developed some designed evaluation functions for the search. Experimental results with various domains (e.g., Home Search, Trip Booking) with GPT-3.5 and GPT-4 show the effectiveness of the proposed method named ToolChain*.

**Strengths:**

- A firm contribution of the reasoning framework using LLMs inspired by the MCTS and search algorithm.
- Extensive experimental evaluations show the effectiveness of the concept.

**Weaknesses:**

- Posiblly handcrafted the two important components (g(n) and h(n) in the A* search) for the problem.

**Questions:**

- Although the evaluated methods and experiments seem solid and interesting to the readers, I’m curious about the designing part of the two important evaluators, $g(n)$ and $h(n)$.
    - As we know, the naive A* algorithm has some theoretical discussions (e.g., admissible heuristic and optimality in the search problems). However, in the current status of the ToolChain*, the idea (evaluating the node with some function) is from the A*, but no theoretical discussions (e.g., the effectiveness of search) have arisen from the design of $f(n) = g(n) + h(n)$. Do you have any insights? Are there any reasons to adopt the current form of functions? Tried different ones? Although the ablation study in Table 2 is useful, particularly restricted to the current form of g and h, we could have further options here.

---

> ### Author Response · Authors · 2023-11-19
> **Initial Response to Reviewer tHXf**
>
> We appreciate Reviewer tHXf for considering our work as a firm contribution with extensive experiments. Your suggestions and comments are very helpful for us in improving presentation quality and clarifications with wider impacts. Please find the corresponding responses below:
>
> **[Weakness 1 & Question 1] Questions about the design process of two important evaluators, g(n) and h(n) in the A$^\*$ search. Further studies exploring different forms of cost functions could provide additional insights beyond existing ablation studies.**
>
> We appreciate your insightful question regarding the design of our cost functions $g(n)$ and $h(n)$ in ToolChain*. Similar to most of the recent advances [1][2][3][4][5] in LLM agents and LLM evaluation, ToolChain* is also based on empirical findings. While lacking systematic theoretical proof, ToolChain* in designing $g(n)$ and $h(n)$ is rooted in empirical findings rather than relying solely on handcrafted functions. To validate the advantage of our proposed value function, we explored various cost function formulations, including self-evaluation prompting LLMs to generate verbalized scores [4][6][7], and vote scores in 'tree-of-thoughts' [3]. Our findings indicated that these scoring methods were not consistently accurate. We identified two key factors guiding our heuristic function design: (1) the black-box nature of state-of-the-art LLMs offering limited information for score formulation, and (2) the observation that similar tasks often share similar logical structures in their solution plans. Driven by the two motivations, we developed heuristic scores to maximize the use of additional information derived from past experiences.
> | Model | Home Search | Trip Booking | Google Sheets | Virtual Home | Average |
> |------------|:-----------:|:------------:|:-------------:|:------------:|:------------:|
> | ToolChain* (Self-Evaluation) | 84.0 | 83.3 | 49.9 | 21.5 | 59.7 |
> | ToolChain* (ToT-Vote) | 82.0 | 81.7 | 53.4 | 21.0 | 59.6 |
> | ToolChain* | 93.0 | 90.8 | 61.4 | 28.6 | 68.5 |
>
> The results demonstrate that a combination of LLM-generated scores and heuristic scores yields superior performance. In the initial stages, due to limited data in the long-term memory, the model predominantly relies on the LLM-generated scores for generating plans. As the long-term memory expands, the heuristic functions increase in accuracy, effectively regularizing and refining the LLM-generated scores.
>
> Furthermore, we have conducted explorations to utilize additional information from open-sourced LLMs in order to formulate a more precise cost function. These explorations include: (1) action likelihood, calculated as the product of token log probabilities within an action; (2) answer probability, determined by the token probability of 'yes' in response to the question 'Is this action step correct?'; and (3) a reward-style probability, which applies weight decay based on the length of tokens in the actions and the total number of actions in a plan. Given the constraints of computational resources and limited time available during the rebuttal session, we conducted additional experimental results using LLaMA-2 (7B) for additional insights:
> | Model | Virtual Home |
> |------------|:-----------:|
> | ToolChain* (action likelihood) | 25.7 |
> | ToolChain* (answer probability) | 26.1 |
> | ToolChain* (reward function) | 27.4 |
> | ToolChain* | 28.7 |
>
> The results indicate that our proposed cost functions outperform other value functions from open-source LLMs. We greatly appreciate your constructive feedback and will include further exploration of incorporating open-sourced LLM information into cost function computation in our future work.
>
> **[Reference]**
>
> [1] Yao et al. “ReAct: Synergizing Reasoning and Acting in Language Models.” ICLR 2023.
>
> [2] Sun et al. “AdaPlanner: Adaptive Planning from Feedback with Language Models.” NeurIPS 2023.
>
> [3] Yao et al. “Tree of thoughts: Deliberate problem solving with large language models.” NeurIPS 2023.
>
> [4] Hao et al. “Reasoning with language model is planning with world model.” EMNLP 2023.
>
> [5] Lu et al. “Chameleon: Plug-and-play compositional reasoning with large language models.” NeurIPS 2023.
>
> [6] Xie et al. “Self-Evaluation Guided Beam Search for Reasoning.” NeurIPS 2023.
>
> [7] Xiong et al. “Can LLMs Express Their Uncertainty? An Empirical Evaluation of Confidence Elicitation in LLMs.” arXiv 2023.

---

> > ### Comment · Reviewer_tHXf · 2023-11-20
> > **Thank you for your response**
> >
> > I appreciate additional results and responses.
> >
> > Using an open-source LLM (i.e., LLaMA-2) is interesting for the academic society. Due to the good writing of the paper, I can understand the findings and contributions of the authors, and to be honest, the results are interesting enough. However, I have some concerns about the balance between technical findings, theoretical justification, and empirical findings for the top-tier conferences, as learning value functions is an essential topic for this field (to evaluate states).
> >
> > Judging from the response and other review discussions, I'll increase my score.

---

> ### Author Response · Authors · 2023-11-20
> **Thank you Reviewer tHXf**
>
> Thank you so much for your thoughtful and constructive feedback, as well as your plan to raise the score. We greatly appreciate your confirmation on the findings and contributions of our work, and the understanding that our results are interesting enough.

---

### Official Review · Reviewer_52p8 · 2023-11-01

**Soundness:** 3 good
**Presentation:** 3 good
**Contribution:** 3 good
**Rating:** 8
**Confidence:** 3

**Summary:**

The paper proposes ToolChain*, a best-first search algorithm for improving the performance of LLM-based agents on sequential decision-making problems. The main contributions are algorithmic and empirical. Specifically, the paper proposes a set of heuristic cost functions leveraging a dataset of solved plans to explore the search space more efficiently. Experiments are conducted on the ToolBench and GSM8K datasets. The results show that ToolChain* outperforms or matches strong recent baselines on these tasks.

===========

UPDATE: I thank the authors for their detailed responses to the reviewers comments. After reading the other reviews and comments, I'm revising my score upwards. The technical details are clearer now and it seems like there's enough here to warrant acceptance. I encourage the authors to incorporate the reviewers feedback into the final version of the paper and make the paper as standalone and reproducible as possible.

===========

**Strengths:**

+ The paper tackles an interesting and important research question with potentially large impact.

+ The paper proposes a simple and intuitively clear best-first search approach to construct an improved LLM-based planning agent. The heuristic function seems novel with potential for broad use across tasks.

+ The proposed method outperforms a number of strong baselines. The experiments include ablations, computational costs and other forms of analyses making it easier to evaluate. The experimental section and appendix has a good amount of detail, aiding reproducibility.

+ The paper is reasonably well-written. The use of illustrative examples made it very easy to understand.

**Weaknesses:**

- Some of the implementation details of the four components of the heuristic function are not clearly explained. These are central to evaluating the algorithmic contributions of the paper and reproducibility so a clearer description would be very useful. I found it particularly difficult to understand the implementation details of the "imagination score". See the questions for details.

- Although the experiments have a good amount of detail including ablation experiments, the paper does not clearly identify the main sources of performance improvement. For example, how much does a better scoring function matter compared with a better LLM or search procedure? How does performance depend on the prompt and other hyper-parameters of the algorithm (heuristic score implementation, dataset size and quality, etc.)? What happens if we use non-LLM versions of the scoring function? The paper mentions the importance of the cost function ("this efficiency gain may stem from the proposed superior cost function") but does not investigate deeper. As a result, it becomes difficult to clearly identify the overall impact and broader utility of ToolChain*, especially compared to similar search-based ideas proposed in Tree-of-Thought.

**Questions:**

- Please explain the details of the Imagination Score in more detail. For example, what does "imagine more concrete future steps" mean? Where is the "imagined plan" generated in Algorithm 1? (I think in the cost calculation $f(n)$ but not sure). How many extra calls to the LLM?

- Is the LCS calculation in $g_{t,1}$ over the sequence of API actions ($a_i \in \mathcal{A}$) alone or does it include additional information about the task or problem in the text string? Do the API actions include the parameters or just the method names?

- How much does overall performance vary with the prompt and the other hyper-parameters of the algorithm (dataset, heuristic score implementation, etc.)?

- How much performance comes from the use of the LLM in different parts of the algorithm vs classical (non-LLM) best-first search? Specifically, how is performance impacted by non-LLM expansion with LLMs only used for node evaluation? Is it possible to construct a search-based baseline which **doesn't** use any LLMs but otherwise incorporates the same intuition (long-term memory, heuristic scoring, etc.)? If yes, how might it perform?
  - In Appendix D.5, is it possible to characterize the computational costs of ToolChain* in more detail? For example, what are the the LLM costs in terms of the number of calls and generated tokens? Dollar cost? Heuristic computational costs? Environment (domain) "simulation" costs associated with the API calls, if any?

---

> ### Author Response · Authors · 2023-11-19
> **Initial Response to Reviewer 52p8 -- PART I**
>
> We appreciate Reviewer 52p8 for considering our work as interesting and important with a potentially broad impact. Your suggestions for clarification and improvement are constructive. Please find the responses below:
>
> **[Weakness 1] Some of the implementation details of the four components of the heuristic function are not clearly explained. These are central to evaluating the algorithmic contributions of the paper and reproducibility so a clearer description would be very useful. I found it particularly difficult to understand the implementation details of the "imagination score". See the questions for details.**
>
> Details of the imagination score are explained in Section 3.3 of our original paper. For further clarifications and an illustrative example, we kindly invite you to refer to our response to Question 1.
>
> **[Weaknesses 2.1] Although the experiments have a good amount of detail including ablation experiments, the paper does not clearly identify the main sources of performance improvement. For example, how much does a better scoring function matter compared with a better LLM or search procedure?**
>
> We would like to emphasize that the improvement of our approach over the state-of-the-art approaches (i.e., baselines) [1][2][3][4] is not from the capability of a better LLM. This has been validated by the comparison between our approach and baselines using exactly the same LLM in Table 1. Instead, the improvement comes from (a) the scoring function design and (b) the search procedure. Although using a more advanced LLM may lead to improved performance, in our comparisons, we always keep using the same LLMs for our approach and baselines in Table 1.
>
> To further understand the improvement by (a) the scoring function design and (b) the search procedure, as well as how significant the improvement is compared to the improvement by a better LLM, we have conducted a more thorough ablation study on ToolChain*. This extensive ablation study includes different base LLM models (GPT-3.5, GPT-4), scoring functions (ToT-Value, ToT-Vote, LLM scores, our proposed cost function $f(n)$), and search procedures (ToolChain*, ToT):
> | Row | Model | Home Search | Trip Booking | Google Sheets | Virtual Home | Average |
> |------------|------------|:-----------:|:------------:|:-------------:|:------------:|:------------:|
> | 1 | ToolChain* (GPT-4, $f(n)$ in Section 3.1) | +5.0 | +6.7 | +7.2 | +5.9 | +6.2 |
> | 2 | **ToolChain$^\*$ (GPT-3.5, $f(n)$ in Section 3.1)** | **93.0** | **90.8** | **61.4** | **28.6** | **68.5** |
> | 3 | ToolChain* (GPT-3.5, ToT-Value [3]) | -9.0 | -7.5 | -11.5 | -7.1 | -8.8 |
> | 4 | ToolChain* (GPT-3.5, ToT-Vote [3]) | -11.0 | -9.1 | -8.0 | -7.6 | -8.9 |
> | 5 | ToolChain* (GPT-3.5, LLM scores) | -7.0 | -5.0 | -5.7 | -6.0 | -5.9 |
> | 6 | ToT [3] (GPT-3.5, $f(n)$ in Section 3.1) | -8.0 | - 4.1 | -9.5 | -8.9 | -7.6 |
> | 7 | ToT [3] (GPT-3.5, ToT-Vote) | -10.0 | -7.5 | -12.8 | -6.8 | -9.3 |
>
> In this table above, $f(n)$ indicates our cost function design in Section 3.1, while ToT-Value and ToT-Vote are two value functions utilized in ToT [3].
>
> Our main contributions lie in the development of a more effective and efficient planning algorithm for LLM agents to address complicated real-world problems. We summarized the following observations from the table above: (1) It is important to note that the design of the scoring function is an integral component of a search procedure. While it is challenging to isolate and compare their individual contributions, our experiments indicate that the scoring function design has slightly more impact than the search algorithm itself. In addition, when compared with baselines under the same base LLMs (GPT-3.5 or GPT-4), both our proposed searching procedures and scoring functions have contributed to model performance (see details in Table 1). (2) ToolChain* shows better performance with GPT-4 than GPT-3.5, suggesting that employing a more advanced base LLM could lead to performance improvement, while this improvement is considered moderate compared to the improvement by our searching procedures and scoring functions.

---

> > ### Author Response · Authors · 2023-11-19
> > **Initial Response to Reviewer 52p8 -- PART II**
> >
> > **[Weakness 2.2] How does performance depend on the prompt and other hyper-parameters of the algorithm (heuristic score implementation, dataset size and quality, etc.)?**
> >
> > Regarding the prompt, to mitigate the potential impact of prompts on model performance, we have consistently used the prompts [5] provided by the datasets across all baselines and ToolChain* to ensure a fair comparison.
> >
> > Regarding the heuristic score, the only two hyper-parameters in ToolChain* are $\alpha$ and $\beta$ in Eq.(1) and Eq.(2). We set $\alpha=\beta=0.5$ by default, aiming for an equal contribution from both the non-heuristic and heuristic functions to the final cost function within the algorithm. To further explore the impact of these hyperparameters, we further experimented with varying $\alpha$ and $\beta$ on virtual home and home search datasets:
> > | Virtual Home | 0 | 0.25 | 0.5 | 0.75 | 1 |
> > |------------|:-----------:|:------------:|:-------------:|:------------:|:------------:|
> > | $\alpha=0.5$, altering $\beta$ | 22.6 | 26.5 | 28.6 | 27.8 | 25.3 |
> > | $\beta=0.5$, altering $\alpha$ | 23.0 | 25.7 | 28.6 | 27.0 | 24.9 |
> >
> > | Home Search | 0 | 0.25 | 0.5 | 0.75 | 1 |
> > |------------|:-----------:|:------------:|:-------------:|:------------:|:------------:|
> > | $\alpha=0.5$, altering $\beta$ | 91.0 | 93.0 | 93.0 | 90.0 | 84.0 |
> > | $\beta=0.5$, altering $\alpha$ | 88.0 | 90.0 | 93.0 | 87.0 | 85.0 |
> >
> > From the experiments, we observe that ToolChain* is relatively stable across different $\alpha$ and $\beta$. For tasks with a smaller and simpler action space (e.g., Home Search), the accuracy of LLM-based non-heuristic functions is usually sufficient, resulting in minimal impact from varying  $\alpha$ and $\beta$. Conversely, in scenarios with larger and more complex action spaces, such as Virtual Home, heuristic functions provide additional information to regularize the non-heuristic ones. Consequently, smaller $\alpha$ and $\beta$, which reduce the contribution of heuristic functions, lead to a noticeable decline in performance. It is important to note that while changing the hyperparameters, our proposed methods still consistently outperform all baselines in the majority of cases.
> >
> > Regarding dataset size and quality, we strictly follow the evaluation protocol and use the same dataset size and quality [5].
> >
> > **[Weakness 2.3] What happens if we use non-LLM versions of the scoring function?**
> >
> > The main contribution of our paper lies in the development of an accurate and efficient planning algorithm for LLM agents, which inherently requires the involvement of LLM scoring functions. LLM agents have demonstrated strong planning and reasoning capability in complicated problem-solving [1][2][3][4], with the advantage of few-shot generalization abilities over different tasks. Correspondingly, LLM-based scores can naturally evaluate the actions via only prompting without additional training. On the contrary, non-LLM scoring functions usually necessitate additional training and can fail to precisely evaluate the effectiveness of actions, thereby not fully unlocking the potential for generalization abilities inherent in LLM agents. Thus, the LLM scoring function has been widely used in most recent LLM agents [3][4][6]. Requiring additional training data, non-LLM versions of score functions were not practical within our evaluation settings and would likely pose similar challenges to all LLM-based agent baselines.
> >
> > To validate the above, we conducted additional experiments to show the importance of LLM scoring functions when using LLM agents  (i.e., heuristic scores outlined in Sections 3.2 and 3.3): We conducted additional experiments to evaluate the model performance where a ToolChain* variant was reliant solely on non-LLM scoring functions (i.e., heuristic scores):
> > | Model | Home Search | Trip Booking | Google Sheets | Virtual Home | Average |
> > |------------|:-----------:|:------------:|:-------------:|:------------:|:------------:|
> > | ToolChain* variant (non-LLM scores) | 52.0 | 38.5 | 40.0 | 19.7 | 37.6 |
> > | ToolChain* | 93.0 | 90.8 | 61.4 | 28.6 | 68.5 |
> >
> > The above results indicate that in the absence of LLM-based cost functions (i.e., self-consistency sampling and LLM imagination scores), the model shows a significant performance drop.

---

> ### Author Response · Authors · 2023-11-19
> **Initial Response to Reviewer 52p8 -- PART III**
>
> **[Weakness 2.4] The paper mentions the importance of the cost function ("this efficiency gain may stem from the proposed superior cost function") but does not investigate deeper. As a result, it becomes difficult to clearly identify the overall impact and broader utility of ToolChain$^\*$, especially compared to similar search-based ideas proposed in Tree-of-Thought.**
>
> To elucidate its importance, we refer to our experimental results in Table 2 and additional experiments addressing Weaknesses 2.1. These experiments demonstrate that alternative cost function designs (by comparing row 2 with rows 3,4,5 in additional experiments in Weaknesses 2.1) or the removal of any component from the cost function (by comparing all results in Table 2) results in a notable average performance decrease of 7.9% and 5.0%, respectively. Moreover, these adjustments to the cost functions in ToolChain* lead to an approximate increase of 1.52x in computation time on average. These empirical findings highlight the significance of the cost function in ToolChain* for plan quality and efficiency.
>
> In addition to the empirical results above, the overall impact and broader utility of our paper can be summarized as follows: (1) We propose ToolChain*, a novel A*-like tree search algorithm, to develop autonomous LLM-based agents for complex planning and reasoning tasks; (2) ToolChain* formulates the action space as a decision tree, effectively mitigating error propagation and expanding search space; (3) We leverage a combination of LLM-based functions and heuristic functions to efficiently navigate the agents in the action space. (4) Our extensive experiments demonstrate the effectiveness and efficiency of ToolChain* in tool usage across five distinct datasets including both planning and reasoning tasks (e.g., home searching, trip booking, mathematics, etc.).
>
> Using exactly the same score function, we compare ToolChain* with the tree-of-thoughts (ToT) approach [3], which relies on depth-first and breadth-first search methods. We summarize the advantages of ToolChain* as follows: (1) ToolChain* is based on A* search, utilizing future cost functions for goal-directed node prioritization. This estimation results in enhanced performance and efficiency compared to the ToT, as demonstrated in Table 1 and Figure 5; (2) ToolChain* significantly reduces the number of nodes that need exploration. While ToT-BFS explores all nodes at a given depth before moving to the next level and ToT-DFS potentially follows irrelevant paths, ToolChain* focuses only on the most promising paths through A* search; (3) ToolChain* is designed for more complex, real-world applications, demonstrating its broader utility in LLM agent tool use and reasoning scenarios.
>
> **[Question 1] Please explain the details of the Imagination Score in more detail. For example, what does "imagine more concrete future steps" mean? Where is the "imagined plan" generated in Algorithm 1? (I think in the cost calculation f(n) but not sure). How many extra calls to the LLM?**
>
> The introduction of the imagination score is available in Section 3.3. We would like to provide further clarifications on the future steps and imagined plans with a detailed example. Consider a current plan $p=(a_0,a_1,\cdots,a_t)$. The future score $h(\cdot)$ aims to estimate the potential future cost required to complete the plan and reach the target. The imagination score is a component $h_{t,2}(\cdot)$ of the future score $h(\cdot)$. To calculate $h_{t,2}(\cdot)$, we initially prompt LLMs to imagine subsequent steps to the current plan, resulting in an imagined plan $p_{im}=(a_{t+1},a_{t+2},\cdots,a_T)$. We then calculate the proportion of current steps present in the imagined plan as the imagination score: $h_{t,2}(n)=t/T$. The imagined plan is generated in the cost function calculation $f(n)$ in Algorithm 1.
>
> For example, assume that we have a current plan *“Agent.WalkTo(dining_room) $\to$ Agent.WalkTo(food) $\to$ Agent.Find(food) $\to$ Agent.Grab(food)”*. We first prompt LLMs to use their imagination to complete the plan, resulting in  *“Agent.WalkTo(dining_room) $\to$ Agent.WalkTo(food) $\to$ Agent.Find(food) $\to$ Agent.Grab(food) $\to$ Agent.Find(table) $\to$ Agent.Put(food, table)”*. We then use the proportion of current steps in the imagined plan as the imagination score, where $h_{t,2}(n)=4/6=0.67$, leading to an imagination-based future cost of $1-h_{t,2}(n)=0.33$.
>
> Concerning the additional API calls, each action requires two separate calls to calculate both the self-consistency sampling score and the LLM imagination score. Consequently, by leveraging the total API calls noted in our response to Question 5, we estimate the number of additional API calls as approximately 53.7 per question, which is similar to the number of extra API calls in other tree search-based methods [3][4].

---

> ### Author Response · Authors · 2023-11-19
> **Initial Response to Reviewer 52p8 -- PART IV**
>
> **[Question 2] Is the LCS calculation in $g_{t,1}$ over the sequence of API actions ($a_i\in\mathcal{A}$) alone or does it include additional information about the task or problem in the text string? Do the API actions include the parameters or just the method names?**
>
> We apologize for the potential confusion. We only leverage the sequence of API actions in our calculations of the LCS calculation $g_{t,1}$. The API functions used in the LCS calculation include both the API function name and the parameters. We will add further clarifications in the updated manuscript.
>
> **[Question 3] How much does overall performance vary with the prompt and the other hyper-parameters of the algorithm (dataset, heuristic score implementation, etc.)?**
>
> For additional experimental results and clarifications, we kindly invite the reviewer to refer to our response in Weakness 2.2 for more details.
>
> **[Question 4] How much performance comes from the use of the LLM in different parts of the algorithm vs classical (non-LLM) best-first search? Specifically, how is performance impacted by non-LLM expansion with LLMs only used for node evaluation? Is it possible to construct a search-based baseline which doesn't use any LLMs but otherwise incorporates the same intuition (long-term memory, heuristic scoring, etc.)? If yes, how might it perform?**
>
> LLM agents have demonstrated strong planning and reasoning capabilities in complicated problem-solving [1][2][3][4][6]. Following the most recent research in this line, our primary scope is to develop a planning method for LLM agents to enhance solution quality and search efficiency. Considering the very recent advances [1][2][3][4][6], approaches involving non-LLM best-first search may not lead to an optimized design, and thus not considered in our current research scope: To incorporate non-LLM best-first search (non-LLM agent) into ToolChain*, we need to consider all potential actions as candidates at each step. Given the extensive action space formed by API functions and their parameters, this approach would necessitate evaluating thousands of nodes at every step, leading to impracticality and inefficiency. In contrast, LLM agents utilize their inherent reasoning and planning capabilities to narrow down to the most promising potential next steps, allowing us to apply our proposed cost function to a more manageable set of options. This significantly reduces the need to exhaustively traverse the entire action space at each step. To assess the feasibility of a non-LLM approach, we experimented with randomly expanding 50 potential actions at each step (where >50 actions are less impractical due to the aforementioned efficiency issue), while keeping the cost function unchanged from the LLM agent settings:
> | Model | Home Search | Trip Booking | Google Sheets | Virtual Home | Average |
> |------------|:-----------:|:------------:|:-------------:|:------------:|:------------:|
> | ToolChain* | 93.0 | 90.8 | 61.4 | 28.6 | 68.5 |
> | Non-LLM best-first search ($f(n)$ in Section 3.1) | 0.0 | 0.0 | 12.0 | 5.2 | 4.3 |
>
> These results offer insights into the comparative performance of LLM-based and non-LLM agents in the context of ToolChain*, highlighting the significant contributions of LLM agents to the efficiency and effectiveness of our method. We observe significantly lower performance (i.e., 0) by the non-LLM agent, because the non-LLM agent struggles to perform effectively when choosing the best one from the numerous available actions, with the evaluations being enabled by our LLM-based score functions. Interestingly, the non-LLM best-first search method demonstrates marginally superior performance on Google Sheets across other datasets, primarily due to the inclusion of single-step solutions in its test set. Other traditional non-LLM A* approaches, as referenced in [7], typically involve training for one-step expansion using additional annotated data, which is not always available in the evaluation sets considering the huge amount of actions.

---

> > ### Author Response · Authors · 2023-11-19
> > **Initial Response to Reviewer 52p8 -- PART V**
> >
> > **[Question 5] In Appendix D.5, is it possible to characterize the computational costs of ToolChain$^\*$ in more detail? For example, what are the LLM costs in terms of the number of calls and generated tokens? Dollar cost? Heuristic computational costs? Environment (domain) "simulation" costs associated with the API calls, if any?**
> >
> > We provide more detailed information regarding the computational costs of ToolChain* (GPT-3.5) across our datasets:
> > | Costs | Expansion | Heuristic Functions | LLM-based Functions |
> > |------------|:-----------:|:------------:|:-------------:|
> > | # LLM Calls per Question| 16.44 | 0 | 107.31 |
> > | $ Prompt Cost (GPT-3.5) per Dataset | 7.22 | 0 | 5.53 |
> > | $ Completion Cost (GPT-3.5) per Dataset | 8.23 | 0 | 9.25 |
> >
> > It is important to note that the heuristic functions in ToolChain* do not require additional LLM calls. Instead, they rely on a long-term memory that stores solution plans. As for the costs related to the environment or domain simulations, these are primarily included in the prompts for expansion. This aspect accounts for approximately $7.22. Similar budgetary considerations apply to very recent advances [1][2][3] that also rely on experiments utilizing OpenAI API calls. Additionally, it is worth noting that the cost of using gpt-3.5-turbo is progressively decreasing within months, making it more affordable in the future. We hope this detailed breakdown helps elucidate the computational costs involved in operating ToolChain*.
> >
> > **[Reference]**
> >
> > [1] Yao et al. “ReAct: Synergizing Reasoning and Acting in Language Models.” ICLR 2023.
> >
> > [2] Sun et al. “AdaPlanner: Adaptive Planning from Feedback with Language Models.” NeurIPS 2023.
> >
> > [3] Yao et al. “Tree of thoughts: Deliberate problem solving with large language models.” NeurIPS 2023.
> >
> > [4] Hao et al. “Reasoning with language model is planning with world model.” EMNLP 2023.
> >
> > [5] Xu et al. “On the Tool Manipulation Capability of Open-source Large Language Models.” arXiv 2023.
> >
> > [6] Xie et al. “Self-Evaluation Guided Beam Search for Reasoning.” NeurIPS 2023.
> >
> > [7] Chen et al. “Retro*: learning retrosynthetic planning with neural guided A* search.” ICML 2020.

---

> ### Author Response · Authors · 2023-11-21
> **Follow-up Response to Reviewer 52p8**
>
> Dear Reviewer 52p8,
>
> We sincerely appreciate your valuable and constructive feedback. We would like to kindly remind you that the author/reviewer discussion phase concludes on November 23rd. We hope our responses and additional experiments have addressed your concerns and improved the paper’s quality. If you have any further suggestions or comments, please feel free to share them. We are looking forward to a constructive discussion during the rebuttal phase.
>
> Best Regards,
> The Authors

---

> > ### Comment · Area_Chair_GZVK · 2023-12-05
> > **From AC.**
> >
> > Reviewer 52p8: if possible, can you comment whether the rebuttal has addressed your concerns?

---

### Official Review · Reviewer_gdT8 · 2023-11-09

**Soundness:** 4 excellent
**Presentation:** 4 excellent
**Contribution:** 4 excellent
**Rating:** 8
**Confidence:** 3

**Summary:**

This paper proposes ToolChain*, an efficient tree search-based planning algorithm to augment large language models (LLMs) with external tools for complex real-world planning and reasoning tasks.  This paper provides very good insight that formulates the action space as a search tree, where each node is an API function call. This mitigates error propagation and expands the search space compared to linear reasoning in open-/closed-loop systems. It uses an A*-like algorithm to search the tree, prioritizing branches using a task-specific cost function with cumulative cost g(n) and future cost h(n). This balances exploration and exploitation for efficient search.

**Strengths:**

1) Novel formulation of LLM planning as tree search using A* search algorithm is intuitive and elegant. Allows structured exploration of expansive action space.
2) Task-specific cost functions g(n) and h(n) provide a principled way to guide search and prioritize promising branches.
3) Strong empirical results demonstrating improved success rate and efficiency over competitive baselines on diverse tasks.

**Weaknesses:**

1) Cost functions rely on heuristic components like long-term memory, self-consistency sampling, and LLM imagination which may not always be accurate.
2) Still trails open-loop systems in efficiency, albeit outperforming other tree search methods. There is a tradeoff between search depth and solution quality.
3) Limited analysis on how the approach generalizes to even more expansive action spaces and longer planning horizons.
However, I think that the above weaknesses are some hard open questions in NLP with LLM.

**Questions:**

1) How robust are the cost functions g(n) and h(n) for entirely new tasks where long-term memory/heuristics may not be available?
2) Could incremental search algorithms like IDA* further improve efficiency over A* search used here?
3) For very large action spaces, is it possible to focus the search on high-level plans first before expanding API details? Hierarchical search?
4) How well would ToolChain* generalize if the action space grows 10x or 100x in size? At what point would efficiency degrade?
5) Beyond API functions, how can ToolChain* be extended to more general symbolic action spaces?

---

> ### Author Response · Authors · 2023-11-19
> **Initial Response to Reviewer gdT8 -- PART I**
>
> We sincerely appreciate Reviewer gdT8 for considering our work as novel and promising with solid empirical results. Your suggestions are decent insights for us to revise the submission draft. Please find the responses below:
>
> **[Weakness 1] Cost functions rely on heuristic components like long-term memory, self-consistency sampling, and LLM imagination which may not always be accurate.**
>
> We recognize that individual heuristic components of our cost functions, such as (1) long-term memory, (2) self-consistency sampling, and (3) LLM imagination, might not always yield perfectly accurate results. Similar to recent works [1][2][3] in the field of LLM agents, our work leverages heuristic functions to evaluate actions during the search for solutions. In this paper, we innovatively identify three intrinsically complementary components and further discover that a combination of these components enhances model performance compared to using any single component in isolation (see Table 2 for empirical details). This improvement stems from each component providing a unique perspective in evaluating the nodes within the search tree. For example, in scenarios with limited plans in long-term memory, the other two components (self-consistency sampling and LLM imagination) become more influential in shaping the cumulative function $f(n)$ and the future score $g(n)$. Conversely, as more plans accumulate in the long-term memory, the heuristic scores gain reliability and effectively regularize the scores generated by LLMs.
>
> **[Weakness 2] Still trails open-loop systems in efficiency, albeit outperforming other tree search methods. There is a tradeoff between search depth and solution quality.**
>
> We agree with the reviewer's point about the efficiency trade-offs of our method in comparison to open-loop systems. As acknowledged in Appendix A, our method, while effective, does not yet match the efficiency of open-loop systems. Our ToolChain* aims to strike an optimal balance between efficiency and solution quality. While open-loop systems are more efficient by directly generating the entire plan, they often compromise on plan correctness due to limited exploration in the potential action space. To fundamentally address the limited exploration issue of open-loop systems, similar to recent Tree-of-Thought (ToT) [1] and Monte Carlo Tree Search (MCTS) [2], our approach is based on a tree search system with inevitable additional costs, resulting in lower efficiency. When compared with recent state-of-the-art tree-search systems [1][2], Toolchain* stands out by achieving the best efficiency (as discussed in Section 4.2). Enhancing model efficiency without significantly compromising performance remains a key direction for our future work.
>
> **[Weakness 3] Limited analysis on how the approach generalizes to even more expansive action spaces and longer planning horizons. However, I think that the above weaknesses are some hard open questions in NLP with LLM.**
>
> We acknowledge the importance of further analysis in more expansive action spaces and longer planning horizons. Besides Table 5 in Appendix C.2 (page 15), we provide more details of our tested environments:
> |            | Home Search | Trip Booking | Google Sheets | Virtual Home | Average |
> |------------|:-----------:|:------------:|:-------------:|:------------:|:-------:|
> | # Tools | 15 | 20 | 108 | 40 | 46 |
> | Avg. # APIs / Solution | 7.5 | 13.4 | 6.1 | 13.4 | 10.1 |
>
> The # TooIs represents the number of different tools in the environment. The Avg. # APIs/Solution represents the average number of different API functions called in each solution. For our evaluated environments, while the number of tools may appear limited, the combination of these tools with flexible parameters significantly expands the action space, which is comparable to ToolLLM [4], the largest known dataset (concurrent to our work) in the tool-use domain. Considering the multi-tool scenarios only compose tools within the same category, the ToolLLM dataset is of a similar scale to ours, averaging around 70 different tools in each category. In terms of solution length, the ToolLLM dataset features solutions with 6 to 15 actions, a range comparable to our datasets. Notably, some solutions in our virtual home dataset extend to approximately 36 actions, indicating a more complex environment.

---

> > ### Author Response · Authors · 2023-11-19
> > **Initial Response to Reviewer gdT8 -- PART II**
> >
> > **[Question 1] How robust are the cost functions g(n) and h(n) for entirely new tasks where long-term memory/heuristics may not be available?**
> >
> > The ToolBench datasets [5] we utilized ensure that all test tasks differ from the seed data in the long-term memory (i.e. where long-term memory/heuristics is not available). ToolBench [5] also introduces 'API complexity' as a metric to measure the difficulty in generalizing to unseen API combinations and non-default argument values. A higher API complexity score indicates that the test data solutions are less familiar to the models. Our selected datasets exhibit the highest API complexity relative to other datasets in ToolBench. We list the API complexity with the average performance gain of ToolChain* over the most competitive baseline in each task:
> > |            | Home Search | Trip Booking | Google Sheets | Virtual Home |
> > |------------|:-----------:|:------------:|:-------------:|:------------:|
> > | API Complexity | 7.3 | 11.1 | 8.4 | 12.3 |
> > | Avg. Performance Gain | 2.0 | 1.7 | 0.2 | 3.7|
> >
> > The above results indicate that ToolChain* demonstrates robust generalization across various scenarios, regardless of the number of unseen API combinations (i.e., API complexity) in the test solutions.
> >
> > **[Question 2] Could incremental search algorithms like IDA$^\*$ further improve efficiency over A$^\*$ search used here?**
> >
> > To the best of our knowledge, ToolChain* novelly represents the very first combination of LLM agent and A* search to achieve a balance between model efficiency and plan quality. We acknowledge that more advanced search algorithms like IDA* could potentially improve model performance and efficiency. However, IDA* may not be ideally suited for tool-use scenarios. First, as a combination of depth-first and A* searches, IDA* is particularly effective in memory-constraint environments with a vast number of both explored and unexplored nodes [6]. In tool-use cases, where we deploy LLMs as agents, the necessity to traverse every possible next step is reduced. LLMs’ planning capabilities allow us to prune irrelevant actions, keeping the number of nodes to be stored at a manageable level. Second, given the high branching factor typical in our scenarios, the efficiency of IDA* could be compromised as it might revisit the same nodes multiple times across different iterations [6]. In future work, we will incorporate ToolChain* with more advanced search algorithms to enhance both efficiency and plan quality.
> >
> > **[Question 3]  For very large action spaces, is it possible to focus the search on high-level plans first before expanding API details? Hierarchical search?**
> >
> > As our main contribution, ToolChain* novelly represents the very first combination of LLM agent and A* search, striking a balance between model efficiency and the quality of planning. Hierarchical search has been leveraged in LLM agents for planning and problem-solving, as evidenced by [7][8][9]. Similarly, we can incorporate hierarchical search with ToolChain* as a potential improvement. Inspired by hierarchical reinforcement learning, we can decompose the problem into sub-tasks, like existing approaches [10], and propose a high-level agent to search among the sub-tasks. Subsequently, a low-level agent would focus on detailed API function calls within the context of these identified sub-tasks by a high-level agent. Both search agents could utilize the underlying principles and mechanisms of ToolChain*.
> >
> > **[Question 4] How well would ToolChain$^\*$ generalize if the action space grows 10x or 100x in size? At what point would efficiency degrade?**
> >
> > We conducted comparisons across datasets featuring varying magnitudes of action space sizes. From Figure 5(a), we notice that the action space size has a less significant impact on model efficiency compared to the average plan length. This is because, during the expansion stage of planning, LLM agents inherently prune the options based on their reasoning and planning abilities. Thus, they do not need to traverse all potential next steps but rather focus on the most promising ones. Our cost function is designed to rank these pruned actions. However, as plan lengths increase, the search space expands exponentially. When the length of plans grows 10x or 100x, the efficiency of ToolChain* may start to degrade. To the best of our knowledge, there are currently no widely used environments with an action space 10x or 100x larger than our evaluation datasets. In addition, it is important to note that the challenge of a growing search space due to longer plans also affects other baselines, keeping the efficiency comparison relatively consistent (see details in Figure 5).

---

> > > ### Author Response · Authors · 2023-11-19
> > > **Initial Response to Reviewer gdT8 -- PART III**
> > >
> > > **[Question 5] Beyond API functions, how can ToolChain$^\*$ be extended to more general symbolic action spaces?**
> > >
> > > We conducted experiments beyond API functions in Section 4.3. Specifically, we evaluated ToolChain* on mathematical reasoning tasks, where arithmetic computations can be considered symbolic actions. Given that general symbolic actions lack the strict formats of API functions, we employed a pre-trained Natural Language Inference (NLI) model to determine if two steps correspond to the same action. By integrating this approach into the ToolChain* framework, we demonstrate its generalization capabilities for broader symbolic action spaces.
> > >
> > > **[Reference]**
> > >
> > > [1] Yao et al. “Tree of thoughts: Deliberate problem solving with large language models.” NeurIPS 2023.
> > >
> > > [2] Hao et al. “Reasoning with language model is planning with world model.” EMNLP 2023.
> > >
> > > [3] Xie et al. “Self-Evaluation Guided Beam Search for Reasoning.” NeurIPS 2023.
> > >
> > > [4] Qin et al. “ToolLLM: Facilitating large language models to master 16000+ real-world APIs.” arXiv 2023.
> > >
> > > [5] Xu et al. “On the Tool Manipulation Capability of Open-source Large Language Models.” arXiv 2023.
> > >
> > > [6] Korf et al. “Time complexity of iterative-deepening-A∗.” Artificial Intelligence 2001.
> > >
> > > [7] Sun et al. “AdaPlanner: Adaptive Planning from Feedback with Language Models.” NeurIPS 2023.
> > >
> > > [8] Wang et al. “Voyager: An open-ended embodied agent with large language models.” FMDM Workshop at NeurIPS 2023.
> > >
> > > [9] Wang et al. “Describe, explain, plan and select: Interactive planning with large language models enables open-world multi-task agents.” NeurIPS 2023.
> > >
> > > [10] Zhou et al. “Least-to-most prompting enables complex reasoning in large language models.” ICLR 2023.

---

> ### Comment · Reviewer_gdT8 · 2023-11-21
> **Thank you for your reply!**
>
> Thank you for your reply! The authors have addressed my primary concerns effectively.

---

> > ### Author Response · Authors · 2023-11-21
> > **Thank you Reviewer gdT8**
> >
> > Thank you very much for taking the time to review our rebuttal and offering insightful feedback. We will update our paper with the additional results and discussions.

---

### Comment · Area_Chair_GZVK · 2023-11-17
**From AC**

Dear Authors,

Would it be possible for you to answer the questions asked by reviewers?

---

> ### Author Response · Authors · 2023-11-17
> **General Response to AC and All Reviewers**
>
> We express our gratitude to all reviewers for their valuable and constructive feedback. In response to the concerns raised, we have summarized our main modifications as follows:
> - We performed additional experiments to benchmark our approach against various alternative cost function designs;
> - We included hyperparameter studies of $\alpha$ and $\beta$ for model robustness;
> - We conducted comparisons with additional variants of non-LLM cost functions and non-LLM agents.
>
> More details are explained in the individual response for each reviewer. We thank the reviewers again and look forward to further suggestions or discussions.

---

### Author Response · Authors · 2023-11-22
**A Gentle Reminder**

Dear Reviewers,

Thank you again for your constructive and valuable feedback! We would like to kindly remind reviewers that the author/reviewer discussion phase ends today, November 23rd. We sincerely hope our responses and additional experiments have addressed your concerns and improved the paper’s quality. If you have any further comments or suggestions, please don't hesitate to share them. We look forward to engaging in a constructive discussion during the rebuttal phase. We wish you a Happy Thanksgiving with your families!

Best,
Authors

---

### Meta-Review · Area_Chair_GZVK · 2023-12-07

**Metareview:**

The paper uses a version of the A star algorithm for planning. The planning problem is to find a sequence of API calls for an LLM-based agent to execute, given a certain task. Most of the paper focuses on how to define heuristic functions used by A star to make it competitive against MCTS.

Strengths:
- a simple, classic approach outperforms strong baselines, including MCTS
- problem setting interesting
- paper is well-written
- as far as I can tell, this application of A star is novel

Weaknesses:
- A star relies on heuristics, which means the approach may not generalise that well to novel tasks / LLMs (having said that, the existing  experiments demonstrate broad enough applicability for the paper to be above the bar)
- Efficiency does not match open-loop systems
- Ablations to get to the bottom of where the performance comes form would be nice (although experiments are understandably costly with LLMs)

**Justification For Why Not Higher Score:**

The approach taken by the paper is somewhat incremental (A star is a very well known method). While the heuristics used by A star are new and the paper is well-executed, I do not see anything truly outstanding about the work.

**Justification For Why Not Lower Score:**

Strong empirical results on an important problem,

---

### Decision · Program_Chairs · 2024-01-16

Accept (poster)